



# Influence of Temperature and Humidity on Contrail Formation Regions in EMAC: A Spring Case Study

Patrick Peter[1,2], Sigrun Matthes[1], Christine Frömming[1], Patrick Jöckel[1], Luca Bugliaro[1], Andreas Giez[3], Martina Krämer[4,5], and Volker Grewe[1,2]

[1]Deutsches Zentrum für Luft- und Raumfahrt, Institut für Physik der Atmosphäre, Oberpfaffenhofen, Germany
[2]TU Delft, Faculty for Aerospace Engineering, The Netherlands
[3]Deutsches Zentrum für Luft- und Raumfahrt, Flugexperimente, Oberpfaffenhofen, Germany
[4]Forschungszentrum Jülich GmbH, IEK-7, Jülich, Germany
[5]Institute for Atmospheric Physics (IPA), Johannes Gutenberg University, Mainz, Germany

**Correspondence:** Patrick Peter (patrick.peter@dlr.de)

**Abstract.** While carbon dioxide emissions from aviation often dominate climate change discussions, the significant impact of non-$CO_2$ effects like contrails and contrail-cirrus must not be overlooked, particularly for the mitigation of climate effects. This study evaluates key atmospheric parameters influencing contrail formation, specifically temperature and humidity, using various model setups of a general circulation model (GCM) with different vertical resolutions and two nudging methods for specified dynamics setups. Comparing simulation results with reanalysis data for March and April 2014 reveals a systematic cold bias in mean temperatures across all altitudes and latitudes, particularly in the mid-latitudes where the bias is about 3-5 K, unless mean temperature nudging is applied. In the upper-troposphere/lower stratosphere, the humidity of the nudged GCM simulations shows a wet bias, while a dry bias is observed at lower altitudes. These biases result in overestimated regions for contrail formation in GCM simulations compared to reanalysis data. A point-by-point comparison along flown trajectories with measurement aircraft data shows similar biases. Exploring relative humidity over ice ($RH_{ice}$) threshold values for identifying ice-supersaturation regions provides insights into the risks of false alarms for contrail formation, together with information on hit rates. Accepting a false alarm rate of 16% results in a hit rate of about 40% ($RH_{ice}$ threshold 99%), while aiming for an 80% hit rate increases the false alarm rate to at least 35% ($RH_{ice}$ threshold 91-94%). A comprehensive one-day case study, involving aircraft-based observations and satellite data, confirms contrail detection in regions identified as potential contrail coverage areas by the GCM.

## 1 Introduction

Estimates of the effective radiative forcing of aviation from 1940 to 2018 indicate that a significant part (best estimate 2/3) stems from non-$CO_2$ effects, including contrails, $NO_x$ and $H_2O$ emissions (Lee et al., 2021). However, this estimated 2/3 contribution is a global and annual average and varies significantly between individual flights (Grewe et al., 2014; Dahlmann et al., 2021; Teoh et al., 2022). Hence, operational mitigation options exist to avoid regions that are more climate sensitive than others. These regions are largely affected by the formation of strongly warming contrails or an over-proportional production of





ozone from $NO_x$ emissions. Methods are required to identify these climate sensitive regions and enable their forecast within numerical weather prediction models.

The ECHAM/MESSy Atmospheric Chemistry (EMAC) model was utilised to calculate climate change functions (CCFs)
(Grewe et al., 2014; Frömming et al., 2021). These CCFs are a 4-dimensional measure of the global climate impact of individual local aviation emissions in dependency of the location, altitude, and time of emission, as well as the prevailing meteorological conditions at the time of emission. By identifying regions which are particularly sensitive to aviation emissions, the CCFs represent the basis for climate-optimized weather dependent aircraft routing (Matthes et al., 2020; Lührs et al., 2018; Grewe et al., 2014). As discussed by Grewe et al. (2017), the implementation of climate change functions for climate-optimized flight
planning is only reasonable if the CCFs and their corresponding mitigation potential are evaluated in a consistent modelling framework. However, validation opportunities are limited. One validation approach has been conducted to test the impact of $NO_x$ emissions on ozone (Yin et al., 2023; Rao et al., 2022).

Hence, we evaluate different EMAC model setups and analyse whether the model is able to reproduce basic meteorological fields from other models or observations. Since contrails have the largest effect on climate-sensitive regions and are therefore
the primary target of air traffic management (Matthes et al., 2020; Lührs et al., 2021), we concentrate on the evaluation of contrail-forming parameters such as temperature and humidity in our study. These parameters are significant in identifying the regions where contrails may form and also influence their life cycle (Schumann, 1996; Kärcher, 2018). For instance, the use of particular temperature thresholds to calculate regions where contrails may form implies that significant differences between models and observations could lead to inaccurate forecasts of such regions and subsequently affect the climate-optimized
aircraft trajectories, leading to an even greater climate response. Therefore, it is crucial that current climate models are evaluated to ensure their accuracy, while at the same time being cautious how model results are used further. The forecast of ice-supersaturation in the atmosphere, which is a crucial factor for the formation of persistent contrails, remains a challenging area in aviation-climate modelling (Gierens et al., 2020; Schumann et al., 2021; Rädel and Shine, 2010; Reutter et al., 2020; Tompkins et al., 2007). Climate models show different levels of humidity biases (Gierens et al., 1997; Brinkop et al., 2015;
Kaufmann et al., 2018; Krüger et al., 2022). The EMAC model employed in this study also shows a temperature bias in the upper troposphere/lower stratosphere (UTLS) region (Stenke et al., 2007; Jöckel et al., 2016) that likely is a consequence of numerical diffusion of water vapour across the tropopause (Stenke et al., 2007) and not uncommon in climate models (Charlesworth et al., 2023). However, no systematic analyses of the influence of model vertical and horizontal resolution or of temperature nudging for "specified dynamics" setups on contrail prediction exist yet. Therefore, we first compare key at-
mospheric parameters for contrail formation and life cycle between different EMAC model setups. Special focus is given on comparing different methodologies of Newtonian relaxation (nudging) towards reanalysis data (specified dynamics (SD) setups).

Second, we compare EMAC simulation results with atmospheric measurements, analysing temperature and humidity data sampled along the trajectories of the High Altitude and Long Range (HALO) research aircraft during the ML-CIRRUS campaign.
Finally, we evaluate satellite images, and compare regions where contrails are visible with predicted contrail forming areas obtained from our SD simulation.



This paper is structured as follows: Section 2 describes the EMAC model and the experimental design (Sect. 2.1), followed by a description of the ECMWF reanalysis data (Sect. 2.3), and atmospheric observations used for the comparison (Sect. 2.4). Section 3 compares temperature and humidity data calculated with different EMAC setups with ECMWF data, while investigating the prediction of contrail formation areas. In Section 4, we compare contrail formation parameters along aircraft trajectories to highlight the discrepancy between simulated and observed quantities. Section 5 presents a more detailed analysis along the trajectory for a selected case study, accompanied by satellite images, and Section 6 discusses our principal findings. In Section 7, we summarise the key points, provide concluding remarks, and suggest aspects for future research.

## 2 Materials and Methods

This subsection outlines the structure of the model and the corresponding simulations (Section 2.1) and explains the different methods to identify atmospheric conditions for the formation of persistent contrails (Section 2.2), followed by a brief summary of the reanalysis data used for the comparison (Section 2.3). In the final subsection, all sources of observational data are illustrated in detail (Section 2.4).

### 2.1 Atmospheric Modelling: The Earth-system model EMAC

Within the framework of the second version of the Modular Earth Submodel System (MESSy2), the EMAC model is utilised to investigate physical processes that are important for the climate effect of contrails. EMAC is a numerical chemistry-climate model simulation system that includes sub-models describing atmospheric processes from the troposphere to the mesosphere and their interactions with the ocean, land and human influences. The fifth generation of the European Centre Hamburg General Circulation Model (ECHAM5; Roeckner et al., 2006) is used as the core atmospheric model. The physical subroutines of the original ECHAM code have been modularised and re-implemented as MESSy sub-models with ongoing development. ECHAM now only retains the spectral transform dynamical core, the flux-form semi-Lagrangian large scale advection scheme (Lin and Rood, 1996), and the Newtonian relaxation-based nudging routines for specified dynamics (SD) model setups (Jöckel et al., 2010; Jöckel et al., 2016).

#### Experiment design and simulation setup

For the present study we applied EMAC (MESSy version 2.55) in a general circulation model (GCM) setup (i.e. without interactive chemistry) with spherical truncation of T42 and T63 (corresponding to quadratic Gaussian grids of approximately 2.8 by 2.8 degrees and 1.9 by 1.9 degrees in latitude and longitude, respectively). We performed simulations with three different vertical resolutions, one with 31 (L31) vertical hybrid pressure layers, another with 41 (L41) vertical hybrid pressure layers, and a third one with 90 (L90) vertical hybrid pressure layers, reaching from the surface up to 10 hPa (L31), 5 hPa (L41), and 0.01 hPa (L90), respectively. The vertical resolution of the 41 layer simulation, similar to that with 90 layers, includes layers of about 20 hPa height in the UTLS region (see Figure 1). While the L31 setup has the fewest vertical layers, it also has the lowest vertical resolution in the UTLS region. Within the L90 setup, there are more layers in the upper stratosphere above



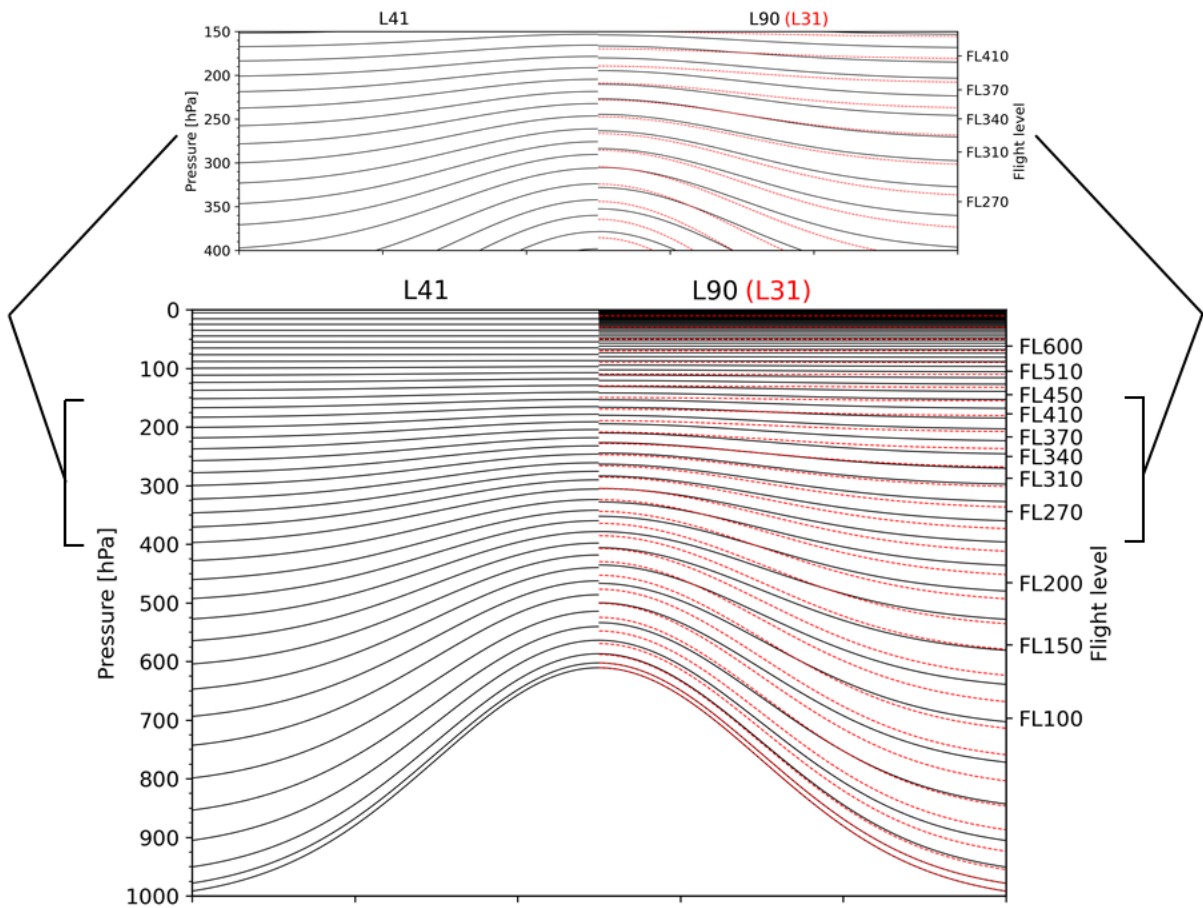

**Figure 1.** Vertical distribution of height levels in EMAC: L41 (left), L31 (right, red), and L90 (right, black) at a ground pressure of 1013 hPa and an idealized orography.

50 hPa compared to the other setups. Due to the absence of these additional layers in the L41 or L31 model setups, various middle and upper stratospheric processes remain poorly represented. Consequently, the upper stratosphere dynamics, such as the Quasi-Biennial Oscillation, are not accurately represented in L41 and L31.

All simulations in this study obtained initial data (monthly averages of January 2013) on humidity and methane from the same L90 pre-simulation RD1SD-base-01. This is a EMAC SD simulation that followed the ref D1-protocol of the Chemistry-Climate Model Initiative 2 (CCMI-2) (Plummer et al., 2021). The simulations were set up with Newtonian relaxation (nudging) towards the ERA5 reanalysis data (Hersbach et al., 2020). Additionally, a simulation with the L41 resolution also underwent nudging towards ERA-Interim reanalysis data for comparison (Dee et al., 2011). The nudging of the ECHAM5 base model is applied in the spectral space for four prognostic variables, namely, divergence, vorticity, temperature, and the logarithm of



**Table 1.** Overview of the EMAC model setups used in this study. For the T42L41 STN setup, two simulations were performed with different nudging data (ERA5 and ERA-Interim, respectively), where STN refers to standard temperature nudging and MTN refers to mean temperature nudging.

| Name[a] | T42L31 STN | T42L31 MTN | T63L31 STN | T42L41* STN | T42L41 MTN | T42L90 STN |
|---|---|---|---|---|---|---|
| Grid cells (lon,lat,lev) | 128 x 64 x 31 | 128 x 64 x 31 | 192 x 96 x 31 | 128 x 64 x 41 | 128 x 64 x 41 | 128 x 64 x 90 |
| Vertical model boundaries [km] | 0–30 | 0–30 | 0–30 | 0–47 | 0–47 | 0–80 |
| Vertical model boundaries [hPa] | Surface–10 | Surface–10 | Surface–10 | Surface–5 | Surface–5 | Surface–0.01 |
| Highest point with max. nudging [hPa] | 126 | 126 | 126 | 100 | 100 | 97 |
| Nudging data | ERA5 | ERA5 | ERA5 | ERA5 (ERAI) | ERA5 | ERA5 |
| Relaxation of global mean temperature | No | Yes | No | No | Yes | No |

(a) T refers to a triangular truncation at a specific wave number (42 or 63), corresponding to a quadratic Gaussian grid of approximately 2.8 by 2.8 degrees and 1.9 by 1.9 degrees in latitude and longitude, respectively, L refers to the number of vertical hybrid pressure levels.

the surface pressure. The corresponding relaxation times for nudging, defined as the time for a model variable to adjust to prescribed data, is 24 hours for temperature and the logarithm of the surface pressure, 6 hours for vorticity, and 48 hours for divergence. The nudging strength is not applied homogeneously in the vertical dimension: the boundary layer and the layers

above 126 hPa (L31), 100 hPa (L41) and 97 hPa (L90) are not nudged, a transition zone with intermediate strength exists in between (see also Jöckel et al., 2016, supplement). Two different nudging settings were used: in two cases, the global mean temperature ("wave zero") was included for the Newtonian relaxations (T42L31 MTN and T42L41 MTN), in all other cases it was omitted. The differences and agreement between the model setups used are listed in Table 1.

All simulations had a time step length of 10 min. As in RD1SD-base-01, the time evolution of the methane tracer at the

lower model boundary was prescribed by Newtonian relaxation towards the data provided by CCMI-2. In the $CH_4$ sub-model (Winterstein and Jöckel, 2021), the stratospheric water vapour contribution from methane oxidation was accounted for by the photolysis calculated online in the photo-chemistry sub-model JVAL (Riede et al., 2009). Two sub-models are of particular relevance in this study: S4D or "sampling in 4 dimensions" (Jöckel et al., 2010) and CONTRAIL (Version 1.0; Frömming et al., 2014). In our study, the CONTRAIL sub-model is used to identify potential areas where contrails and contrail cirrus may

form according to the Schmidt-Appleman Criteria (SAC) (Schumann, 1996). In addition, the potential contrail cirrus coverage is determined, taking into account regions where contrails may persist once formed. Both variables are calculated following Burkhardt et al. (2008). With the S4D sub-model it is possible to sample model data along aircraft flight trajectories online i.e. during the model simulation. In previous studies, a standard bi-linear horizontal interpolation was used to compute the values along these tracks. For this study, we used the recently implemented nearest-neighbour approach, which provides grid-box local



results for temperature and humidity. Vertical interpolation, if required, is linear in pressure altitude in both cases. The position files of the aircraft used in this study are obtained from the initial Midlatitude Cirrus (ML-CIRRUS) campaign flight track data. They have a time resolution of one minute, but sampling only takes place at every model time step (10 min). Subsequently, the data sets were separated into tropospheric ($PV \leq 2.5$) and stratospheric ($PV > 2.5$) data by using the model parameter potential vorticity (PV). The standard PV of 2.5 was selected for the tropopause according to Zängl and Wirth (2002). Based on

the analysis of the data, it is found that 55% of the data points were present in the troposphere ($PV \leq 2.5$), whereas 45% were located in the stratosphere ($PV > 2.5$). Initially, a spin-up simulation was performed separately for each model setup from January 2013 to February 2014, producing output of global data, either instantaneous or temporal averages every 5 hours. The main simulations were then performed from 1 March 2014 to 30 April 2014, with an output interval of 1 hour. All analyses were conducted for the extended North Atlantic Flight Corridor (NAFC, 120°W-20°E), Asia (50°E-160°E), and the Northern

Hemisphere, between 80°N and 20°N, at pressure levels where air traffic takes place. The NAFC is of particular interest, because CCFs have been calculated for this area in the past (Frömming et al., 2021).

## 2.2   Methods for identifying atmospheric conditions to form persistent contrails in global climate models

In this study, we investigate two different approaches to calculate contrail formation areas while we do not distinguish between

linear contrails and contrail cirrus. Firstly, we computed ice-supersaturated regions (ISSR) using the method suggested by Dietmüller et al. (2023). Two criteria are defined to identify ISSRs: The temperature must be below 235 K to separate from mixed-phase regions (Pruppacher et al., 1998), relative humidity with respect to ice ($RH_{ice}$) must exceed 100% (see Yin et al., 2023, their supplementary material on contrail aCCFs). However, when studying the sub-grid-scale variability in the relative humidity field of numerical weather forecast model data, such as ERA5, it is essential to take into account $RH_{ice}$ thresholds

below 100% as demonstrated by Irvine et al. (2014). Dietmüller et al. (2023) explored different relative humidity over ice thresholds for ERA5 and conducted a comparison with MOZAIC data (Petzold et al., 2020) in the European region. The aim was to pre-determine areas where contrails form before assessing the climate effect using algorithmic climate change functions for these identified regions. They found the best agreement between ERA5 and observations is achieved when the $RH_{ice}$ threshold is set to 90%.

Alternatively, the atmospheric ability to form persistent contrails and contrail cirrus in EMAC can be calculated at each time step according to Burkhardt et al. (2008) and Burkhardt and Kärcher (2009). Here, the fraction of a grid box available for contrail coverage is determined by the Schmidt-Appleman criterion, incorporating local temperature and humidity conditions conducive to contrail formation. This fraction is calculated as the difference between the potential coverage of contrails and cirrus clouds combined and the coverage of natural cirrus clouds alone. For our study, we utilise values for the overall propul-

sion efficiency of an aircraft and the fuel combustion heat, which are essential for calculating the SAC, from Schumann et al. (2000). Further details about the calculation method for the potential contrail coverage are given by Frömming et al. (2021) and Grewe et al. (2014).



## 2.3 ECMWF reanalysis data

To assess simulation results fast and efficiently, a comparison with validated model results is useful. Reliable reanalysis data
are required for an objective evaluation. Therefore, we utilised hourly and monthly global operational reanalysis data from
the European Centre for Medium-Range Weather Forecasts (ECMWF) for March and April 2014. For our comparison, we
used the fifth generation of ECMWF reanalysis (ERA5) data with a spatial resolution of approximately 31 km and 137 vertical
hybrid levels reaching from the surface up to 0.01 hPa (about 80 km). At typical flight altitudes around 11.5 km (200 hPa), the
vertical resolution is approximately 300 m (10 hPa). A comprehensive set of monthly mean temperature and specific humidity
ERA5 data, which were aggregated from daily means, were provided by the German Climate Computing Center DKRZ on
the High Performance Computing system Levante. The data was re-mapped on a 0.25° x 0.25° grid and interpolated from the
original ERA5 hybrid model levels to pressure levels. The same procedure was applied to the hourly data of temperature and
specific humidity for 26 March 2014 (Hersbach et al., 2020). As there is no monthly or hourly relative humidity data available
on Levante, we acquired this data from the Copernicus Climate Data Store. For temperatures above 273.15 K, the relative
humidity is calculated based on saturation over water, while for temperatures below 250 K, it is calculated based on saturation
over ice. For temperatures between 250 K and 273.15 K, it is determined by interpolating between the values for ice and water
using a quadratic function. The relative humidity data has 37 interpolated pressure levels and was re-gridded to a regular lat-lon
grid of 0.25°. For our study, we only used relative humidity data between 200 hPa and 350 hPa (Hersbach et al., 2023). All
data sets have global coverage.

## 2.4 Atmospheric observations

Further knowledge on the performance of the model can be gained through comparing model data and observations. While
a global coverage is normally not available for most atmospheric parameters at high resolution, comparing specific areas
can provide valuable insight into the accuracy of model parameters and opportunities for model improvement. In this study,
measurement data from the Midlatitude Cirrus (ML-CIRRUS, Voigt et al., 2017) measurement campaign are compared with
EMAC model results at cruise altitude. Furthermore, satellite measurements were utilised to compare the areas where contrail
formation was predicted in the model with observed contrails.

**Aircraft Experiment ML-CIRRUS**

The ML-CIRRUS campaign, which took place from March 10 to April 16 2014, was one of the first scientific missions to
demonstrate the capabilities of the novel High Altitude and Long Range Research Aircraft (HALO; www.halo.dlr.de). A com-
prehensive overview of the scientific objectives, flight plan, and equipment is provided by Voigt et al. (2017). Over the campaign
period, HALO conducted 16 research flights, which amounted to a total of 88 flight hours. The flights were designed to com-
prehensively characterise mid-latitude cirrus and contrail cirrus using in situ and remote sensing instruments. The ML-CIRRUS
project aimed to enhance our understanding of cirrus cloud formation across varying meteorological conditions (Krämer et al.,
2016; Luebke et al., 2016; Wernli et al., 2016; Urbanek et al., 2017), to refine our estimations of the radiative impact of cirrus



(Krisna et al., 2018), and to assess air traffic impacts on high cloud cover (Schumann et al., 2017; Grewe et al., 2017; Li et al., 2023). The flights covered almost the entire central European region from the northern British coast to Portugal. To fulfil the scientific objectives of the mission, the HALO payload for ML-CIRRUS consisted of the instrumentation for measuring cloud particles, aerosols and trace gases including five different water vapour instruments. For our research, we analysed data collected by the Fast In situ Stratospheric Hygrometer (FISH). This closed-cell Lyman-$\alpha$ photofragment fluorescence hygrometer

has been used on multiple research aircraft for over two decades (Meyer et al., 2015; Schiller et al., 2009). The instrument's operating principle is described in detail by Zöger et al. (1999). FISH is capable of measuring water vapour mixing ratios ranging from 1 to 1000 ppm. The overall uncertainty during ML-CIRRUS was found to be 6% relative with a $\pm0.4$ ppm absolute offset uncertainty (Kaufmann et al., 2018). It is important to note that FISH measures 'total water,' which includes gas phase water as well as evaporated ice crystals. When comparing with gas phase $H_2O$ from the model, only values up to 80% $RH_{ice}$

are used to exclude clouds, as within clouds, $RH_{ice}$ is elevated due to ice particles. Therefore, we substituted all FISH values (with a water vapour mixing ratio > 15 ppmV) where $RH_{ice}$ was larger than 80% with measurements from the SHARC sensor. SHARC (Sophisticated Hygrometer for Atmospheric ReseaRCh) is a tunable diode laser (TDL) hygrometer and is part of the HALO basic data acquisition system (BAHAMAS; Giez et al., 2023). This closed-cell hygrometer uses the absorption line of water vapour at 1.37 $\mu$m. More details about the configuration of the SHARC sensor during the ML-CIRRUS campaign can

be found in Kaufmann et al. (2018). The temperature data used in this study was also measured by BAHAMAS (Giez et al., 2023).

Since the interval of the observation data is 1 sec, the alignment with the 10-minute interval of the model is important. Initially, we computationally derived weighted average mean values for the measurement data through the implementation of a Gaussian filter. We utilized the fact that infinite repetitions of a boxcar filter can be converted into a Gaussian filter (Gans and Gill, 1984)

and therefore performed multiple loops of a centered rolling mean to achieve a weighting of the data. Following this, the data was resampled to 10-minute intervals. In order to focus on contrail formation areas and to remove outliers, observation values were restricted below 240 K, resulting in 308 data point pairs or 51 hours of flight time. The correlation between observational and model data was calculated using a linear regression function.

**Satellite Data**

The SEVIRI instrument aboard the geostationary Meteosat Second Generation (MSG) satellite is used to observe contrails in the region probed by the HALO aircraft during ML-CIRRUS on 26 March 2014. These observations were conducted over the North Atlantic region, specifically near the coast of Ireland. This area is strategically significant as it lies along the major air traffic routes between Europe and the United States, making it a prime location for studying contrail formation and behavior (Sec. 2.4, see also Wang et al. (2023) for a detailed evaluation of this flight). The SEVIRI imager (Schmetz et al., 2002) aboard

the operational MSG satellite MET-10 observes the Earth in 11 spectral channels with a spatial sampling distance of 3 km at the sub-satellite point and a temporal resolution of 15 min. Due to its location in the geostationary orbit above $0°$ E this sensor is well-suited to observe Europe. However, it has limitations, such as being unable to fully capture regions like the Western North Atlantic Ocean. Additionally, the spatial resolution decreases towards the edge of the observed Earth disk, making it



challenging to identify small structures like contrails. In this study, false colour composites (so called ash RGBs) are used in
order to visually spot contrails. These ash RGBs combine three brightness temperatures and temperature differences for the
MSG/SEVIRI channels centred at 8.7, 10.8 and 12.0 $\mu$m. Here, contrails appear as dark blue or black lines and are easy to
identify, so this method is often used for this purpose (e.g. Meijer et al., 2022). Furthermore, the use of purely thermal channels
makes these observations independent of solar illumination such that the ash RGBs can be used during day and night. However,
the moderate resolution of MSG/SEVIRI only allows to distinguish contrails that are 30-60 min old (Vázquez-Navarro et al.,
2015; Gierens and Vázquez-Navarro, 2018). These are persistent contrails which have formed in ice-supersaturated regions and
can live for hours. For the sake of comparison with the model results, the data is remapped from the original satellite projection
to an equal longitude-latitude grid.

## 3 Influence of different vertical model resolutions and nudging techniques on temperature, humidity and ice super-saturation

As the representation of key meteorological parameters is critical for the understanding of contrail formation and life cycle, this
sections identifies differences in the atmospheric distribution of temperature and humidity between EMAC simulations with
various vertical resolutions (L31, L41, L90), horizontal (spectral) resolutions (T42, T63), and different nudging approaches
(standard temperature nudging (STN), or mean temperature nudging (MTN)). Firstly, the simulated temperature and humidity
of various model setups are compared to ERA5 data in three specific regions (NAFC, Asia and the whole northern hemisphere)
for March and April 2014. In order to better understand the local peculiarities, the parameters were examined separately within
20° latitude bands for all selected regions. In the end, this analysis focuses on the temperature and humidity data for 26 March
2014. Here, two distinct methods are used to calculate potential contrail formation regions for this specific day.

### 3.1 Temperature and humidity profile comparison for March and April 2014

Figure 2 illustrates the area weighted mean temperature across six distinct EMAC model setups (see Table 1) for the extended
NAFC between 50 hPa and 400 hPa in comparison with ERA5 for March and April 2014. Notably, our comparison reveals
a cold bias in EMAC simulations with standard temperature nudging compared to ERA5 reanalysis data for the analysed re-
gion. The identified bias for the area-weighted average mean temperature data ranges from 3 K to 5 K between 400 hPa and
50 hPa, with the most significant discrepancy around 200 hPa. Increasing the spectral (horizontal) resolution from T42 to T63
leads to a minimal reduction of the bias, by up to 0.2 K, while changes in vertical resolution have negligible effect. The STN
240 model setup with 41 vertical layers exhibits the strongest deviation from ERA5, being 0.1 K colder than other STN setups up to
100 hPa. The analysis of 20° latitude bands for the NAFC (Figure 4, left) indicates a consistent altitude-dependent temperature
bias across all bands. However, the application of mean temperature nudging (T42L31MTN, T42L41MTN), as explained in
Section 2.1, significantly reduces, as to be expected, the cold bias to less than 0.1 K across all analyzed pressure levels and
bands. In the MTN setup, the bias is only present above 80 hPa, in a region where no nudging is applied. The impact of both
nudging concepts on the temperature difference relative to ERA5 is also evident in the Asian region (see Figure S2a and S4a)



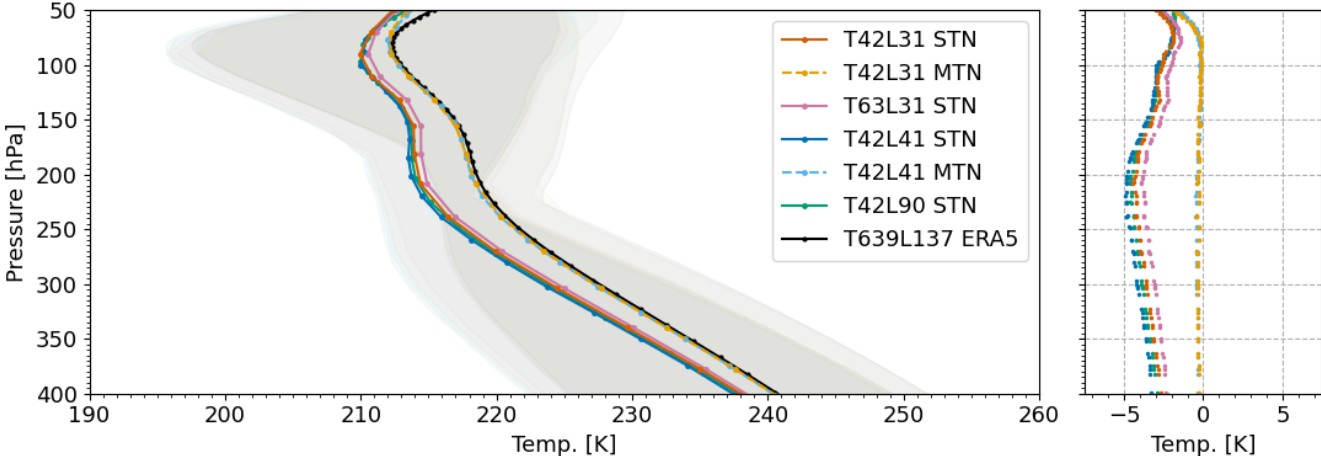

**Figure 2.** Left: Area-weighted mean temperature distribution from EMAC simulations with different vertical resolution (L31: red, L41: blue, L90: green) and ERA5 data (L137: black) for the North Atlantic Flight Corridor. The mean values for each grid box were calculated beforehand by using hourly data spanning from March to April 2014. Nudging approaches: standard (STN, *solid*) and mean (MTN, *dashed*) temperature nudging, with 95th percentiles shaded. Right: Temperature difference between EMAC and ERA5.

and across the entire Northern Hemisphere (see Figure S2b and S4b). Small differences can be found in the Asian region above 80 hPa, where simulations (except L90) exhibit a warm bias (up to 1 K), regardless of the applied nudging method.

The analysis of relative humidity over ice in the extended NAFC during March-April 2014, presented in Figure 3, reveals up to 30% higher mean values at each EMAC grid point compared to ERA5 across four pressure levels (200 hPa, 250 hPa, 300 hPa,
and 350 hPa). The MTN simulations, on average, exhibit closer agreement with ERA5 than the STN simulations, with noticeable differences across pressure levels. The most substantial differences between both nudging methods are observed at 200 hPa, where MTN simulations demonstrate the best agreement with ERA5, deviating by an average of 7.5 to 10%, while the difference to the STN simulations shows a peak between 12.5 to 15% absolute difference. At 250 hPa and 300 hPa, differences between all STN model setups are further reduced, while the difference between EMAC and ERA5 remains consistent
on average. Within each nudging group, variations between simulations with different model levels are minimal, but increasing the spectral (horizontal) resolution to T63 significantly diminishes the difference to ERA5 across all analyzed pressure levels. The variation in distributions between simulations with mean and standard temperature nudging is smaller for the NAFC region than for the entire Northern Hemisphere. In the Northern Hemisphere (Figure S3b), a larger spread in the relative humidity over ice difference between EMAC and ERA5 can be observed for all analyzed simulations and pressure levels, with certain EMAC
grid points indicating lower relative humidity than ERA5. In model setups with standard temperature nudging, the ERA5-EMAC difference increases with altitude, ranging from 12% average difference at 350 hPa to 18% at 200 hPa. In contrast, MTN simulations consistently have their peak values between 7.5 and 15% difference across all pressure levels. In the Asian region (Figure S3a), distributions at 250 hPa and 300 hPa resemble those of the NAFC region and the Northern Hemisphere,





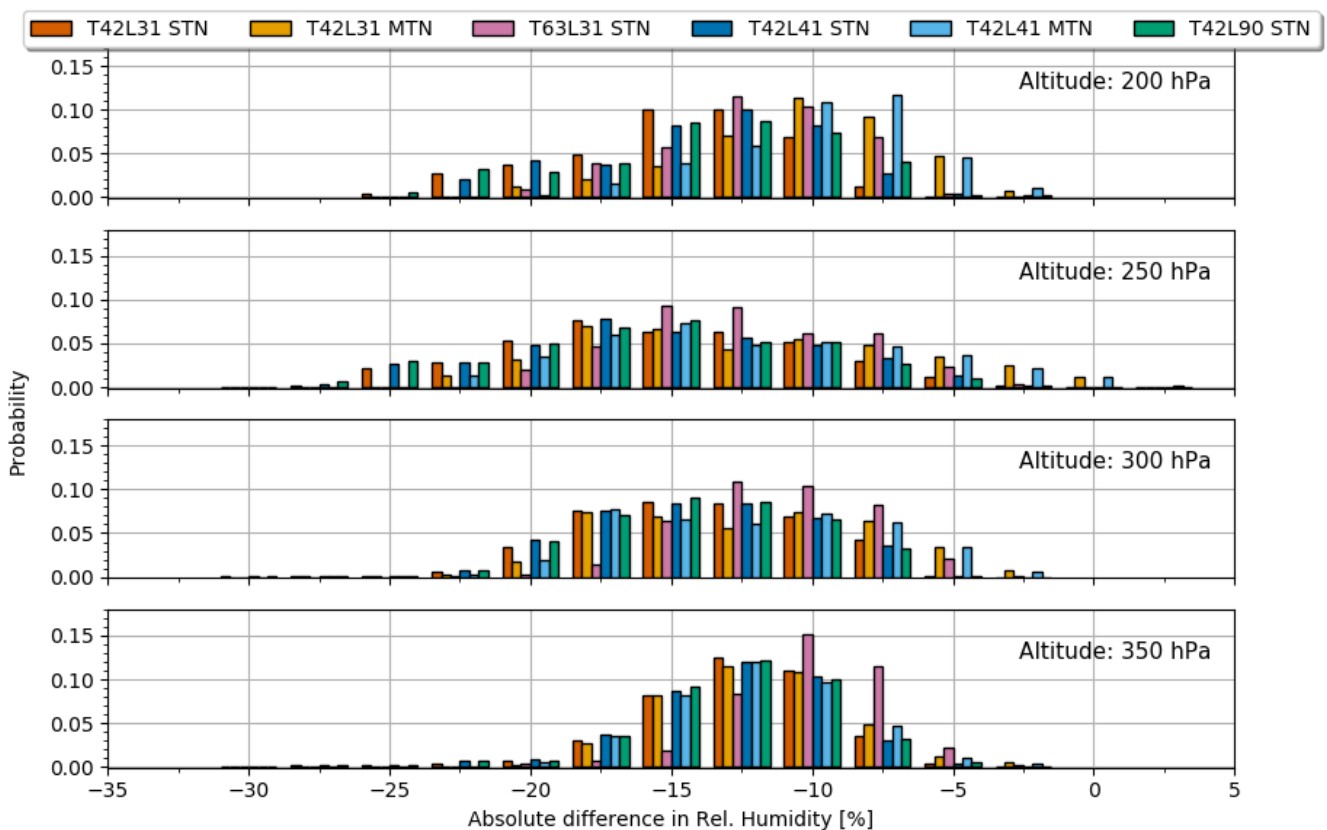

**Figure 3.** Probability density functions illustrating the absolute differences in relative humidity over ice (given in percentage points) between various EMAC model setups and ERA5 data (ERA5 minus EMAC) in the extended NAFC region for March and April 2014. The figure displays the distribution for four vertical pressure levels, starting from 200 hPa at the top and concluding with 350 hPa at the bottom. At each pressure level, the vertical model resolution and nudging method is indicated by different colors: red (L31), dark blue (L41), and green (L90) for standard temperature nudging, and orange (L31) and light blue (L41) for mean temperature nudging. A dataset with higher horizontal resolution (T63) is represented in purple.

while at 200 hPa substantial disparities from ERA5 are evident. STN simulations exhibit an average absolute difference of

22% in relative humidity to ERA5, while the MTN simulations difference peaks at around 19%. At 350 hPa, STN simulations perform even 1%-point better than their MTN counterpart. The altitude-dependent cold bias can partly explain the disparity in relative humidity between ERA5 and EMAC for the STN simulations. However, an analysis of specific humidity values is necessary to understand discrepancies in simulations involving mean temperature nudging.

Subsequently, we examined the vertical distribution of specific humidity along 20° latitude bands for the three selected areas.

In the extended NAFC (Figure 4), an altitude and latitude-dependent humidity bias that differs with the nudging concept is apparent. For MTN simulations, a consistent wet bias is observed at higher altitudes (150 hPa to 250 hPa), significantly differing







**Figure 4.** Temperature and humidity distribution from various EMAC simulations with different vertical resolution (L31:red, L41:blue, L90:green) and ERA5 data with 137 vertical level (black) for the North Atlantic Flight Corridor. Nudging approaches: standard (STN, *solid*) and mean (MTN, *dashed*) temperature nudging. Left: Area weighted mean temperature distribution and difference to ERA5 for all EMAC setups. Right: Area weighted mean humidity mixing ratio distribution and difference to ERA5 for all EMAC setups. The mean values for each grid box were calculated beforehand by using hourly data spanning from March to April 2014. The 95% percentile of each simulation is marked as grey shade.

from ERA5. While differences are minimal in the tropics, a substantial bias exists in the mid-latitudes (40° to 80°). This results in higher relative humidity over ice values for MTN simulations compared to ERA5, despite similar temperatures. In contrast, simulations driven by STN exhibit a dry bias below 250 hPa for all latitude bands compared to ERA5. Additionally, they display a wet bias north of 40° at heights between 150 hPa and 250 hPa. The dry bias at lower levels (350 hPa) compensates for the slight temperature bias, yielding relative humidity over ice values closer to ERA5 when compared to other pressure levels (Figure 3). At 300 hPa, STN simulations reveal a temperature bias increase, while the dry bias remains unchanged,



resulting in a minimal rise in relative humidity over ice difference to ERA5. The temperature bias in STN simulations peaks at 250-200 hPa, accompanied by a wet bias intensifying at higher latitudes. However, the overall bias is reduced on average due

to lower values at lower latitudes. No significant differences are observed between simulations with different vertical resolutions. In the other investigated regions, similar results are found. It is evident that a substantial bias in specific humidity exists in the MTN setup, particularly at higher altitudes and northern latitudes. Nevertheless, the effect on relative humidity weakens at higher altitudes because of the pressure-level-dependent vapour pressure.

### 3.2 Contrail forming areas in different model setups

Our numerical simulations utilizing seven different model setups reveal differences in horizontal size (area) of regions where contrails could form (potential contrail cover) or persist. With both methods outlined in Section 2.2, we first analyze contrail coverage areas for thresholds at 90%, 95%, and 100% $RH_{ice}$ below 235 K at three distinct pressure levels (see Table 2) in a specific synoptic situation (26 March 2014, 12:00 UTC), where research aircraft observations from the ML-CIRRUS campaign are available (see Section 2.4). Reducing the humidity threshold from 100% to 95% and to 90% results in an increase in

contrail coverage areas by a factor of about 2 and 2.5, respectively. Due to the differences in temperature and humidity between the model and ERA5 reanalysis data, all EMAC setups generally exhibited a larger contrail formation region compared to ERA5. Simulations with standard temperature nudging, irrespective of vertical resolution, indicate larger contrail coverage areas for each pressure level and $RH_{ice}$ threshold. Transition to a T63 resolution yielded a marginal decrease in coverage of approximately 1%-point. At 250 hPa, MTN simulations display nearly identical covered areas as ERA5 for the 90% (15-16%

coverage) and 95% (11-13% coverage) thresholds. In contrast, STN simulations show 10% (> $RH_{ice}$ 90%) to 8% (> $RH_{ice}$ 95%) more potential contrail formation areas at this pressure level. At 300 hPa and 350 hPa, MTN simulations overestimate ERA5 results by 5% to 7%, while STN simulations overestimate by 7-16%. However, for the 100% threshold, EMAC MTN results agree with ERA5 at 300 hPa, while they underestimate at 250 hPa by 3% and overestimate at 350 hPa by 3-4%.

The vertical temperature profiles from EMAC and ERA5 on the specified day (see Figure S2c) display a model cold bias of

similar magnitude (3-5 K, peaking around 200-250 hPa) as derived from monthly mean values (Figure 4). While specific humidity differences exhibit similar patterns, relative humidity distributions show even larger discrepancy, including cases where the $RH_{ice}$ of ERA5 exceeds that of EMAC across all setups (see Figure S3c). The findings summarized in Table 2 suggest that both, the cold bias in the standard temperature nudged simulations, and the overestimation of humidity in the simulations including mean temperature nudging in EMAC, significantly affect the estimated areas of ice-supersaturated regions. While

the selection of the nudging method has a significant impact, the number of model levels only affects the results at specific pressure levels.

The influence of different vertical model resolutions and nudging methods on the second contrail formation area identification approach, as described in Section 2.2, is also presented in Table 2 for the NAFC. While the area-weighted mean potential contrail coverage (AWM PotCov) for grid boxes with a potential contrail coverage above zero remains consistent across all

model setups (0.36 ± 0.02 at 250 hPa, 0.35 ± 0.04 at 300 hPa, 0.34 ± 0.03 at 350 hPa), variations appear in the total region covered by these grid boxes. At 250 hPa and 300 hPa, simulations with 31 and 90 vertical levels show a total coverage





**Table 2.** Percentage of areas indicating contrail formation based on the ISSR and PotCov methods at various pressure levels, using different model setups over the NAFC on on 26 March 2014, at 12:00 UTC. For the ISSR method, three relative humidity over ice ($RH_{ice}$) thresholds (90%, 95%, and 100%) are used. Potential contrail cover (PotCov) areas are calculated directly in EMAC and are provided as total (PotCov > 0) and grid box-adjusted (GBA) coverage areas. The GBA coverage is determined by multiplying each grid box area by the potential contrail cover for that box, then dividing the area-weighted sum of all covered areas by the area-weighted sum of the entire area. The area-weighted mean potential contrail coverage (AWM PotCov) is calculated over all grid boxes with potential contrail coverage above zero.

| Region: 120°W 20°E | T42L31 | T42L31 | T63L31 | T42L41 | T42L41 | T42L90 | ERA5 |
|---|---|---|---|---|---|---|---|
| 80°N 20°N | STN | MTN | STN | STN | MTN | STN | |
| **Height: 250 hPa** | | | | | | | |
| Cov. Area (ISSR) | | | | | | | |
| $RH_{ice}$ > 90 / 95 / 100 [%] | 25 / 19 / 7 | 15 / 11 / 4 | 23 / 19 / 8 | 25 / 21 / 7 | 16 / 13 / 4 | 26 / 21 / 9 | 16 / 13 / 7 |
| Cov. Area (PotCov) | | | | | | | |
| GBA (total) [%] | 24 (62) | 22 (59) | 22 (59) | 23 (58) | 21 (53) | 22 (60) | |
| AWM PotCov [frac] | 0.36 | 0.35 | 0.35 | 0.38 | 0.38 | 0.35 | |
| **Height: 300 hPa** | | | | | | | |
| Cov. Area (ISSR) | | | | | | | |
| $RH_{ice}$ > 90 / 95 / 100 [%] | 31 / 25 / 12 | 22 / 17 / 7 | 29 / 23 / 10 | 32 / 27 / 12 | 22 / 18 / 8 | 31 / 26 / 13 | 15 / 13 / 7 |
| Cov. Area (PotCov) | | | | | | | |
| GBA (total) [%] | 20 (62) | 19 (62) | 19 (59) | 22 (57) | 19 (56) | 22 (60) | |
| AWM PotCov [frac] | 0.34 | 0.32 | 0.32 | 0.39 | 0.35 | 0.37 | |
| **Height: 350 hPa** | | | | | | | |
| Cov. Area (ISSR) | | | | | | | |
| $RH_{ice}$ > 90 / 95 / 100 [%] | 20 / 17 / 10 | 15 / 13 / 8 | 19 / 15 / 9 | 21 / 18 / 10 | 15 / 13 / 7 | 21 / 17 / 10 | 9 / 8 / 4 |
| Cov. Area (PotCov) | | | | | | | |
| GBA (total) [%] | 14 (44) | 12 (41) | 13 (44) | 15 (47) | 12 (43) | 14 (47) | |
| AWM PotCov [frac] | 0.34 | 0.33 | 0.32 | 0.34 | 0.32 | 0.33 | |

between 59 and 62%, while the setup with 41 levels shows coverage between 58 and 53%, with the MTN setup being lower than the STN setup. At 350 hPa, both MTN setups show the lowest coverage (41%), while the STN setups range from 44 to 47%. Summing the geographic grid box areas of those with potential contrail coverage greater than zero results is an overestimation of the areas covered with contrails. Therefore, we adjusted the covered area in each grid box with the respective grid box fraction, reducing the potential contrail areas by nearly 60%. Since this method employs the thermodynamic Schmidt-Appleman criterion, temperature and humidity thresholds play a significant role. Variations in these critical parameters account for differences in the resulting covered areas. Furthermore, the vertical resolution appears to have a smaller impact on the areas compared to the previous method. Since the potential contrail parameter is specific to EMAC, direct comparison with ERA5 is





not possible. Therefore, we conducted an inter-comparison between both approaches.

The comparison between contrail formation areas based on the ISSR method and the PotCoV method reveals the best agreement at the 95% $RH_{ice}$ threshold, although discrepancies persist across different pressure levels. Standard nudged simulations show the closest agreement between both methods at 250 hPa, while simulations with mean temperature nudging perform best at 350 hPa. Specifically, at 250 hPa, MTN underestimates, while at 350 hPa, STN overestimates the potential contrail areas by

5-10%. Compared to ERA5, the EMAC PotCov method consistently overestimates the covered areas for all pressure levels and model setups, with discrepancies ranging from 1-12%. Our findings suggest that the PotCov approach for mean temperature nudged EMAC data yields the best agreement with ERA5 ISSR areas. However, even with this approach, EMAC simulations result in larger contrail areas compared to ERA5. Furthermore, aside from the discrepancies caused by different nudging approaches, the number of model levels appears to have a negligible impact.

## 4   Comparison with observations along trajectories (ML-CIRRUS Campaign)

In general, the climate impacts of non-$CO_2$ effects are complex and require detailed knowledge about key atmospheric parameters. Nevertheless, numerous difficulties may arise when modelling parameters such as temperature or humidity using a general circulation model (Jöckel et al., 2016). Consequently, it is advisable to utilize atmospheric measurements to assess model performance in particular regions. Hence, we compared temperature and humidity data on aircraft trajectories obtained

during the ML-CIRRUS campaign with the emulated trajectory data from different model setups. From the measurement data we derive potential contrail formation conditions e.g. ice-supersaturation, that can be compared with EMAC model potential contrail areas and validated with satellite imagery.

**Temperature comparison between EMAC and ML-CIRRUS measurements**

All model setups exhibit a satisfying correlation with the temperature data collected during the ML-CIRRUS campaign, as

illustrated in Fig. 5. This is supported by a strong correlation coefficient (r) between 0.98 and 0.99.

The STN simulations show lower temperatures than the corresponding observations, indicating a constant cold bias of up to 4 K between 200 K and 240 K. The simulations show a slope almost equal to one, consequently producing parallel fitting lines with the central line. The MTN simulations display temperature pairs distributed around the central line with considerable variations up to 1 K for higher temperatures. Besides that, the 308 data pairs suggest small differences between the MTN setup

and the ML-CIRRUS temperature data. Between the simulations with different vertical resolutions, no differences were found in the respective nudging groups. Changing from T42 to T63 horizontal resolution only has a minimal impact and reduces the average bias by a maximum of 0.1 K. A probability density function assessment of only tropospheric and stratospheric data (see Sec 2.1) indicates the presence of a comparable cold bias in both pressure regions, which is significantly mitigated by utilizing the mean temperature nudging method in the simulation (Figure S5a and S5b). The differences between the MTN simulation

results and the measurements can mainly be attributed to the resampling of the observation data and the uncertainties during the measurement process.





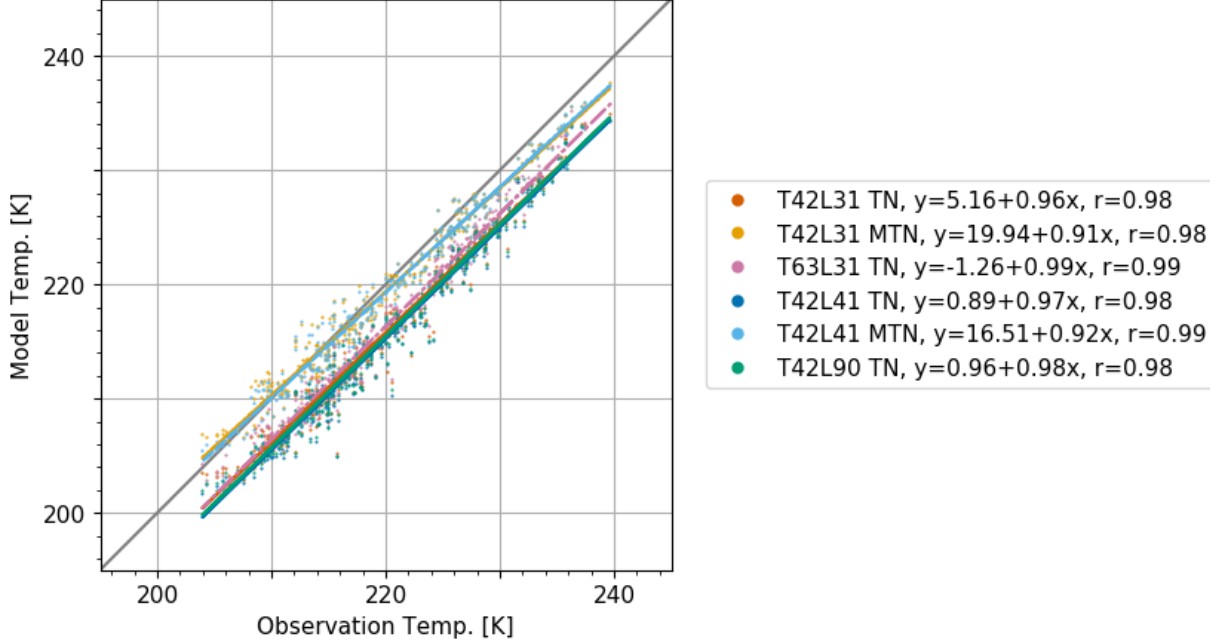

**Figure 5.** Correlation between temperature data measured by the ML-CIRRUS campaign and simulated with various EMAC model setups at different points along the flight path. Next to the color bar, the intercept and slope of the linear regression line for each setup are listed, along with the Pearson correlation coefficient (r).

**Humidity in EMAC compared to ML-CIRRUS measurements**

The analysis of water vapour mixing ratios across various model setups and measurement data, illustrated in Figure 6, reveals a substantial overprediction of low humidity values in the model compared to the observations. These low humidity values

are located in the lower stratosphere and the values are up to six times higher across all model setups. This model "wet bias" for values between 5 to 10 ppmV of the observed water vapour mixing ratio appears with both nudging methods. For high levels of humidity ranging between 100 and 800 ppmV, predominantly present in the upper troposphere, the results from the standard temperature nudging simulations persistently provide lower values than the HALO FISH and SHARC measurements (see Figure 6, left). This "dry bias" is strongly reduced in simulations with mean temperature nudging (see Figure 6, right).

Overall, the measurements indicate a higher number of data points with water vapour mixing ratio values smaller than 5 ppmV compared to the model. The selection of vertical resolution only has a minor impact on the strength of the correlation. In the lower stratosphere, data from T42L41 show the strongest agreement with the ML-CIRRUS data. This results from the fact that the T42L41 model setup has the highest resolution around the tropopause (refer to Section 2.1) in comparison to the other EMAC model setups, which reduces the water vapour diffusion from the troposphere into the stratosphere. For the troposphere,

all simulations produce a similar outcome.





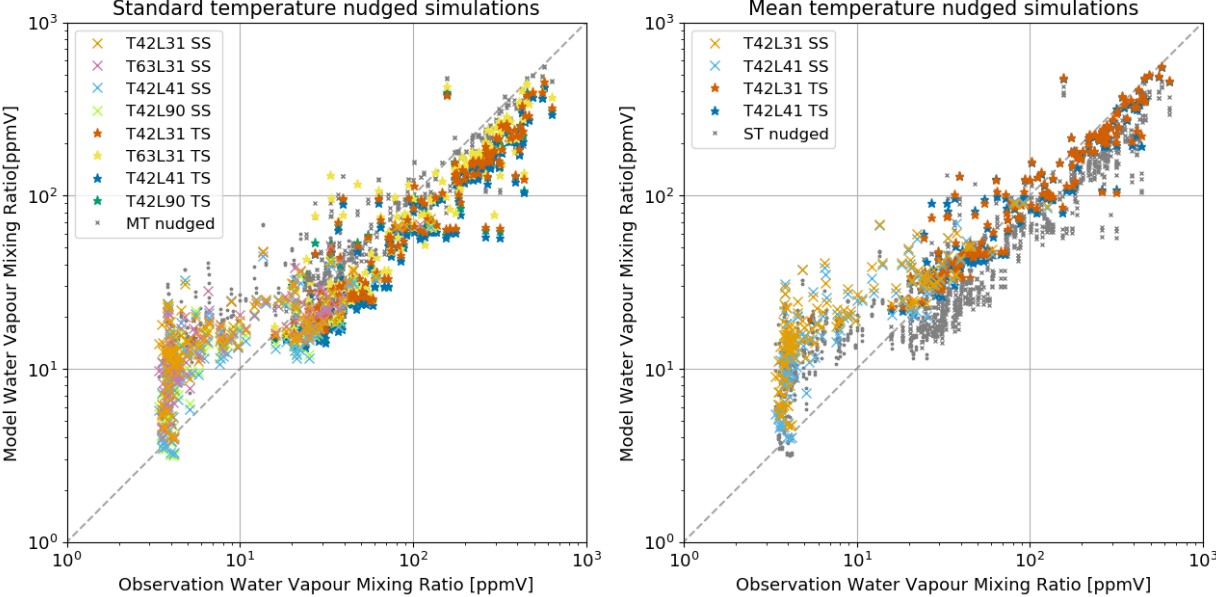

**Figure 6.** Correlation between water vapor mixing ratio measurements from the ML-CIRRUS campaign and results obtained from different EMAC setups. Standard temperature nudged simulations are colored on the left side, while simulations with mean temperature nudging are shown in grey. On the right side, this arrangement is reversed. The respective vertical model resolutions are indicated in red (L31), blue (L41), and green (L90). Stratospheric (SS, crosses) and tropospheric (TS, stars) values are distinguished, with stratospheric values depicted in lighter shades.

**Relative humidity conditions for persistent contrails**

We analyze the point-by-point correlation between measurement data and different model setups for contrails that persist longer than a few minutes and significantly impact the climate, following the approach of Gierens et al. (2020). We focus on ice-supersaturated conditions, where the ambient relative humidity with respect to ice ($RH_{ice}$) exceeds 100%. To achieve this,

we compared $RH_{ice}$ values derived from FISH and SHARC water vapor mixing ratio and temperature measurements (using equations A1-A3, see Appendix) from the ML-CIRRUS campaign with corresponding data from EMAC. Following Dietmüller et al. (2023), we gradually lower the relative humidity threshold for the model data from 100 to 90%. This comparison (refer to Table 3 for the 90% threshold) presents values of a 2 × 2 contingency table with four categories: Y/Y (hits), where both ML-CIRRUS and EMAC data indicate conditions that support contrail persistence; Y/N (misses), where ML-CIRRUS mea-

surements predict contrail persistence but the model does not; N/Y (false alarms), where ML-CIRRUS measurements do not predict contrail persistence but the model does; and N/N (correct negatives), where both ML-CIRRUS and EMAC data show that contrail persistence is not possible.

  Table 3 reveals that for the selected model $RH_{ice}$ threshold (90%), the T42L90 standard nudged model setup achieves the highest hit rate (HR) at 83%, indicating a strong ability to correctly identify contrail conditions. All standard nudged simu-





**Table 3.** Comparison of measured relative humidity with respect to ice derived from ML-CIRRUS data (threshold 100%) and corresponding data from different EMAC setups (threshold 90%). The contingency table shows the correlation between the two data sets, including hits (Y/Y), misses (Y/N), false alarms (N/Y), and correct negatives (N/N). The Equitable Threat Score (ETS) characterizes the agreement between the data sets, considering hits, misses, and false alarms while adjusting for random chance. The hit rate (HR), expressed as a percentage, represents the proportion of contrail conditions correctly identified by the model (Y/Y) relative to the total measured contrail conditions (Y/Y + Y/N). Conversely, the false alarm rate (FAR), also expressed as a percentage, measures the proportion of incorrectly identified contrail conditions by the model (N/Y) relative to the total measured non-contrail conditions (N/Y + N/N) (ref to Ebert (1996)).

| Model setup | Size | Y/Y | Y/N | N/Y | N/N | ETS | HR [%] | FAR [%] |
|---|---|---|---|---|---|---|---|---|
| T42L31 STN | 286 | 23 | 7 | 106 | 150 | 0.08 | 77 | 41 |
| T42L31 MTN | 286 | 21 | 9 | 91 | 165 | 0.08 | 70 | 36 |
| T63L31 STN | 286 | 24 | 6 | 106 | 150 | 0.08 | 80 | 41 |
| T42L41 STN | 286 | 24 | 6 | 108 | 148 | 0.08 | 80 | 42 |
| T42L41 MTN | 286 | 21 | 9 | 99 | 157 | 0.07 | 70 | 39 |
| T42L90 STN | 286 | 25 | 5 | 111 | 145 | 0.08 | 83 | 43 |

lations show high hit rates between 77% and 83%, while the mean temperature nudged simulations only show a hit rate of 70%. Conversely, the MTN simulations exhibit fewer false positives. The T42L31 MTN model achieves a false alarm rate (FAR) of 36%, the lowest in the table, suggesting the fewest incorrect contrail condition identifications. In over one-third of all cases, the model predicts contrail conditions at this threshold that are not confirmed by measurements, irrespective of the vertical resolution or the nudging approach. The calculations were also performed for other $RH_{ice}$ thresholds (Fig. 7). Here,

the trade-off between sensitivity (hit rate) and specificity (false alarm rate) for different model setups and thresholds becomes apparent. Lower thresholds generally result in higher hit rates and false alarm rates. A false alarm rate below 20% could only be achieved for $RH_{ice}$ thresholds over 97%, with a corresponding hit rate below 50%. Aiming for high hit rates above 80% increases consequently the false alarm rate to at least 35%, for $RH_{ice}$ thresholds of 94%.

At high thresholds, the T42L31 MTN setup performs the best. At lower thresholds (90-95%), the T63L31 and T41L90 setups

consistently show higher hit rates than other setups, but they also exhibit higher false alarm rates. The T42L41 setups with standard and mean temperature nudging demonstrate consistent performance with relatively high hit rates and moderate false alarm rates, making them the most balanced model setups. In addition, we also use the Equitable Threat Score (ETS) in Table 3 to characterize the agreement between the data sets. The ETS considers hits, misses, and false alarms while adjusting for random chance, providing a more equitable assessment of forecast performance. The values can be interpreted as follows: ETS

= 1 indicates a perfect forecast (all hits, no misses or false alarms), ETS = 0 indicates the forecast is no better than random chance, and ETS < 0 indicates the forecast is worse than random chance. For a detailed explanation of the ETS, refer to Gierens et al. (2020). In our analysis, the ETS values range from 0.07 to 0.08, indicating a low degree of agreement between the two data sets. This suggests that the relationship between the data sets is mostly random and that this statistical analysis does not confirm a clear correlation, which might be due to the small number of data points.



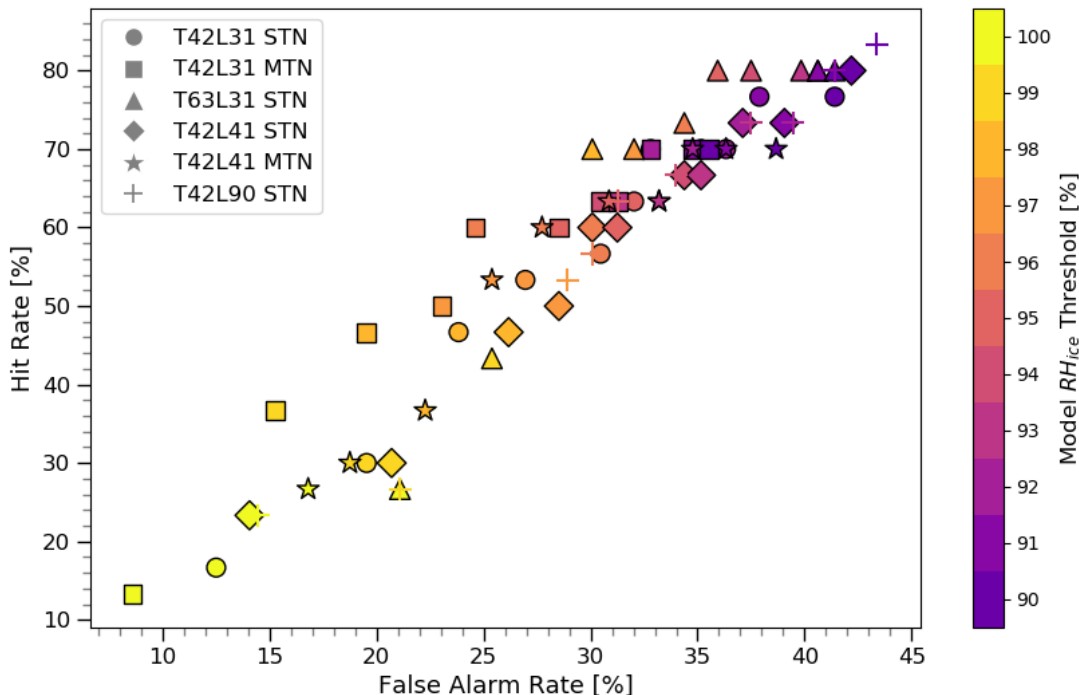

**Figure 7.** Scatter plot illustrating the relationship between the false alarm rate and hit rate across various model setups and $RH_{ice}$ thresholds. Each marker type represents a different dataset: Circles (T42L31 STN), Squares (T42L31 MTN), Triangles (T63L31 STN), Diamonds (T42L41 STN), Stars (T42L41 MTN), and Crosses (T42L90 STN). The color of the markers corresponds to the model $RH_{ice}$ threshold values, with the gradient ranging from purple (90%) to yellow (100%).

## 5 Case study, ML-CIRRUS flight on 26 March 2014

The analysis of all ML-CIRRUS flights provide important information regarding distinctions between model results and observations. However, these findings do not provide insight into the rapid and abrupt changes in flight level and the subsequent alterations in atmospheric conditions. Additionally, the weather conditions must be considered. Thus, we are monitoring the trajectory of a specific flight to acquire a comprehensive understanding. As a case study, 26 March 2014 has been chosen. This day displays intriguing meteorological conditions with a polar cold front outbreak across Europe, causing a subsequent descent of the tropopause height. Temperature, humidity, and potential contrail coverage are compared along the aircraft trajectory for all EMAC setups. Further, the projected areas of possible contrail coverage are compared with satellite imagery.



**Weather situation during the flight**

Following the methodology of REACT4C (Irvine et al., 2012), we have categorised the weather situation over the North
Atlantic on the 26 March 2014. Geopotential height anomalies at 250 hPa indicate minimal correlation with both, the North
Atlantic oscillation index, and the East Atlantic index. Moreover, the jet stream is limited to the western North Atlantic and
exhibits a relatively low intensity. According to Irvine et al. (2012) this results in type W5, a weak confined jet. This type
is the most frequent type in winter time and occurs on average on 26 days per winter. High pressure is positioned above
Scandinavia, while low pressure dominates across Central Europe. Additionally, another low pressure system is located over
Southern France and the Mediterranean Sea. At the southern tip of Greenland, a major low pressure system can be observed,
with a warm and a cold front attached, displaying distinct cloud bands across the North Atlantic in the MSG data. In the nudged
EMAC simulation, the geopotential height at 500 hPa resembles the reanalysis data of the Deutscher Wetterdienst (Berliner
Wetterkarte). The potential contrail coverage (Figure 9) is large where the warm and cold fronts are located, although the
potential contrail coverage (Frömming et al., 2014) is considerably more pronounced around the warm front. These results are
in line with the observations made by Kästner et al. (1999), who noted that contrail formation is more frequent before warm
fronts, before cold fronts, and together with cirrus in a warm conveyor belt.

**S4D sub-model analysis**

A direct analysis along the trajectory provides insight into parameter fluctuations that are not apparent in larger dataset statistics.
Figure 8 illustrates a detailed comparison of the parameters throughout the flight. The analysis indicates that the ML-CIRRUS
flight on the 26th of March 2014 was primarily located in the stratosphere with three descents into the troposphere at 08:30,
09:35, and 10:55 UTC. Each of these crossings of the tropopause lasted roughly 45 minutes and covered pressure differences
from 20 to 100 hPa in depth (a). A significant temperature difference between the observations and the model results in both,
the troposphere and the stratosphere, can be observed for the standard nudged simulations. The discrepancy ranges from 3 to
5 K and is minimally affected by the aircraft's ascent or descent. For the simulation with mean temperature nudging (blue
and orange dashed lines), this cold bias is not present (b). Regarding tropospheric humidity, lower values are found in the
standard nudged simulations compared to the measurement data. However, by applying the MTN method, this bias is largely
reduced. In the stratosphere, the models humidity is up to six times larger than the measurements, regardless of the nudging
method applied. This is particularly noticeable in the first quarter of the flight, but can also be seen whenever the aircraft has
crossed the tropopause into the stratosphere, i.e. for low humidity values. During the final quarter of the flight, the difference
between track pressure and tropopause height is sufficiently large, so that the water vapour mixing ratio values are very low,
below 5 ppmV, and there are no significant differences between model results and observation anymore in terms of water
vapour mixing ratio (c). All model setups indicate a relative humidity over ice of approximately 100% in the troposphere
for the first two dives, which is consistent with the measurement results. However, during the third dive, the observed $RH_{ice}$
are lower and the model-predicted $RH_{ice}$ values exceed the observational data in parts by roughly 20%. This suggests that
the simulated gradient is not steep enough, as the flight is further away from the tropopause than in the dives before. In the





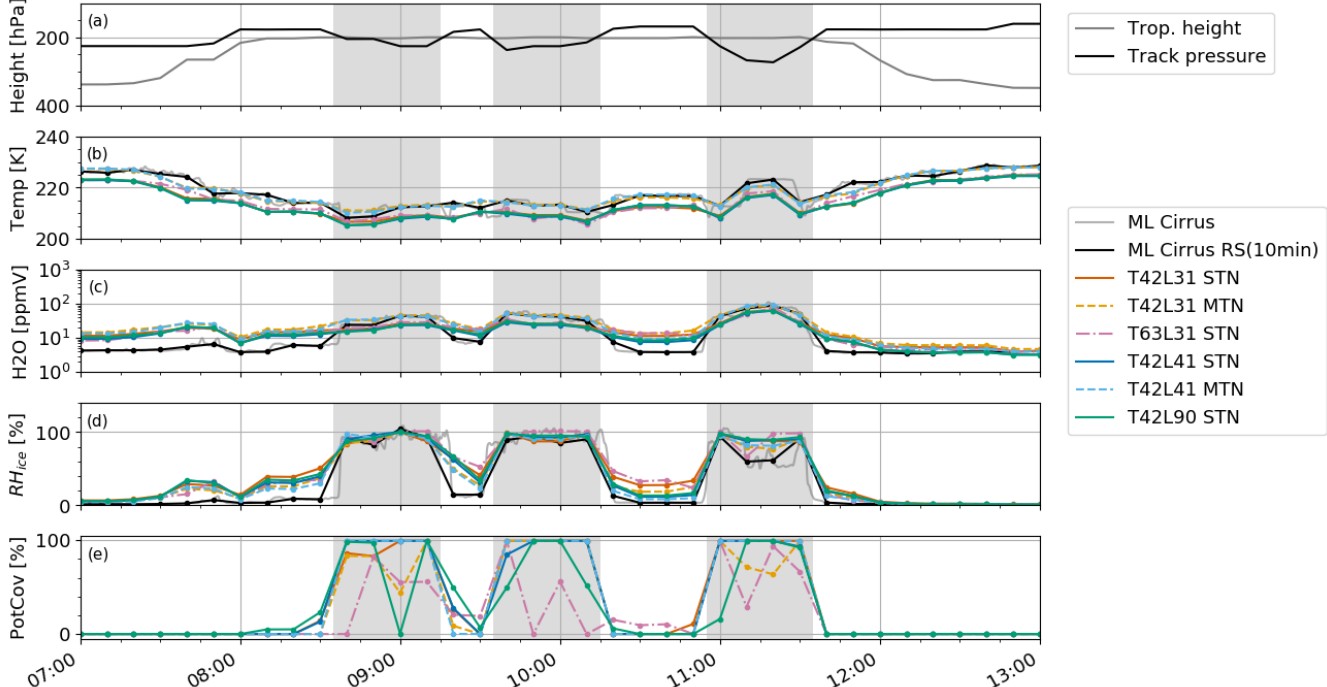

**Figure 8.** Comparison of various atmospheric parameters along the flight trajectory for March 26 2014. (a) Tropopause height and the track pressure height of the HALO aircraft. Comparison of temperature (b), water vapour mixing ratio (c) and relative humidity with respect to ice (d) between results obtained with different EMAC setups and ML-CIRRUS measurements (FISH and BAHAMAS). (e) Shows the predicted contrail coverage in EMAC for the according grid box.

stratosphere, due to the larger mixing ratio of water vapour visible in (c), the model simulates a relative humidity over ice between 10 to 40%, whereas the observational data are nearly zero in this area. Although this may not significantly impact contrail prediction, as it is far from complete saturation, it does require attention. When contrasting the various model setups, the L41 with mean temperature nudging comes closest to the measurement humidity data (c and d). Based on the temperature

and humidity data, contrail formation is expected for the first 2 dives into the troposphere, and partly during the last dive. All model setups predict contrail formation in the troposphere, some also showing a low probability in the lower stratosphere close to the tropopause. The simulation with the finest spectral (horizontal) but lowest vertical resolution, T63L31, predicts the lowest fraction of possible contrail formation in the grid boxes in the troposphere. Applying mean temperature nudging decreases the likelihood of contrail formation for L31 but has no effect for L41. However, the impact of nudging on potential

contrail cover for this specific day is minimal. The impact of the 10-minute output interval for the model data compared to the 1-second observational data is significant, as there is a clear delay in the model data after the aircraft's altitude adjustment. Overall, differences in temperature and humidity, which were also observed in the previous sections, can be determined, if only this particular day is analysed. Comparison along the trajectory provides insight into the significance of the model output



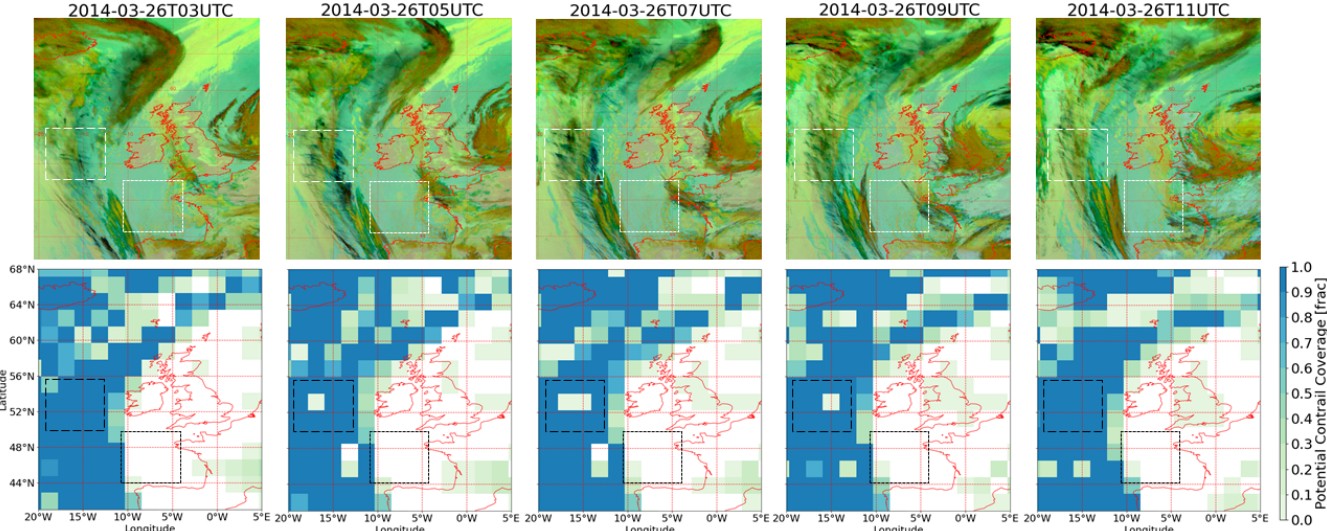

**Figure 9.** Top: The MSG satellite ash RGB from 20°W to 5°E and 68°N to 44°N for 26 March 2014, in 2-hour intervals from 3 UTC (left) to 11 UTC (right). Contrails are visible as dark lines. Bottom: The maximum potential contrail coverage between 300 and 200 hPa (3 model levels) derived from the EMAC L31T63 STN model setup. Dark blue indicates high potential grid box coverage, while light green indicates low coverage. The white and black squares indicate areas with high (–) and no (...) contrail coverage.

and corresponding resampling interval. It should be noted that the models ability to capture all features observed during the steep dives is limited, if the model output interval is too coarse. There is a high level of correspondence between the predicted contrail formation produced by the model and the measured contrail conditions. Between the model configurations, both L41 versions, with or without mean temperature nudging, demonstrate promising results regarding contrail prediction.

**Potential contrail coverage and satellite observations**

The satellite image (top panel in Fig. 9) shows a ridge cloud on 26 March 2014 composed of a cloud band extending from Iceland southwards passing West of Ireland and Great Britain. Contrail lines are best observed in correspondence to the cloud band over a wide range of latitudes starting East of Iceland / North of Scotland and ending East of Portugal, with many contrails East of Scotland and Ireland and less contrails South of Ireland. The spread of these contrail lines is strongly influenced by large-scale synoptic weather patterns. Although the areas where contrails were observed remained consistent throughout the observed time span of 8 hours, a notable increase in intensity occurred at 05 UTC and 07 UTC, coinciding with the period when air traffic from the US to Europe is passing these regions. Of course, contrails can only form where aircraft fly. The maximum potential contrail coverage in the L31T63 STN model indicates the highest value observed between 300 and 200 hPa (bottom panel in Fig. 9). This analysis reveals that the distribution of potential contrail coverage follows the distribution of the ridge cloud in the ash RGB. EMAC indicates the ridge cloud as location of potential contrail formation, which is plausible in comparison to the satellite picture. Additionally, areas where no contrails are predicted, such as south of Ireland, correspond to





regions with no observed satellite contrail coverage. As the ML-CIRRUS campaign conducted measurements east of Ireland on this day, the observed area of contrail formation can also be linked to high relative humidity over ice values. Further regions of high potential contrail coverage, e.g. over South Scandinavia and North of Corsica, cannot be evaluated with these satellite observations, because no contrails are observed here. The region East of England with high potential cloud coverage is also difficult to evaluate, because the few dark lines in the ash RGB in this region (see e.g. 05 UTC) cannot be unambiguously

attributed to contrails.

## 6   Discussion

Temperature and relative humidity are key parameters for determining contrail formation and evolution in numerical modeling studies. This study compares atmospheric temperature and humidity fields from various EMAC model setups with observational data obtained from measurements along aircraft trajectories. We quantify and compare the areas of the atmosphere

conducive to the formation of persistent contrails across different model setups and reanalysis data. Additionally, we explore the differences arising from varying humidity thresholds and methodologies. Our study evaluates key meteorological fields comprising temperature in distinct EMAC model setups, aiming to determine representation within the EMAC T42L41 setup (Frömming et al., 2021) with 41 vertical layers and 2.8° horizontal resolution in the light of evaluating the ability of the model to represent contrail conditions in the Northern Hemispheric extra-tropics. Our analysis reveals a 3 K to 5 K cold bias for the

standard nudged EMAC model temperature compares to both, observations and other models, across the 100-400 hPa range. In the past, general circulation models with a model top at 10 hPa have often been associated with a cold bias problem (Stenke et al., 2007), which we confirm with our analysis. For our study, we find a consistent cold bias in all EMAC simulations, irrespective of vertical or horizontal resolution applied within the setup. Additionally, the bias is found to be equally prominent in the NAFC, Asia and the entire Northern Hemisphere without any geographical distinction based on latitude. The inclu-

sion of global mean temperature ("wave zero") to the temperature nudging (MTN setup) significantly reduces the cold bias in both investigated setups (L31, L41). An extra simulation performed with ERA-Interim based mean temperature nudging for the T42L41 model setup produces nearly identical results to ERA5 (results have not been visualized as they are nearly identical). In the comparison of this MTN setups with ERA5 reanalysis data, there is hardly any cold bias visible between 100 and 400 hPa, independent of the studied region. These findings could have been expected, as the nudging in EMAC was directed towards

ERA5; with the mean temperature nudging technique, temperature data is more aligned with ERA5 data. For the comparison with observations, we notice a good agreement between EMAC model results that utilized mean temperature nudging and the measurement results from the ML-CIRRUS campaign. These findings primarily indicate agreement between ERA5 reanalysis and measurements as expected. However, recent findings have provided new insights into the anti-ice (AI) correction for aircraft measurements (Giez et al., 2023), and indicate a possible cold bias in the temperature data which we want to discuss here.

During the ML-CIRRUS campaign, no AI correction was applied to the measurement data, based on experimental evidence suggesting it was unnecessary. However, recent test campaigns have shown that this does not hold for all flight conditions. The correction is generally negligible, with a value of less than 0.1 K across most flight conditions. However, it becomes significant



at very high altitudes with low speeds, which are not typical flight conditions for the HALO aircraft. The application of the AI correction results in a slightly lower static air temperature, but this correction remains within the specified error margins for temperature measurements. Therefore, the temperatures calculated during the ML-CIRRUS campaign are likely unaffected by the new AI correction.

Similar to results from other studies, model temperature shows a smaller bias when mean temperature nudging is applied. However, such mean temperature nudging has potentially adverse effects on the radiation balance or the hydrological cycle (Jöckel et al., 2016), as no recalibration was performed. This effect becomes apparent when analyzing the specific humidity results. Humidity poses a challenge in climate modelling due to its spatial and temporal variability. It is noteworthy that specific humidity is overestimated in models such as the ECMWF's Integrated Forecasting System IFS, while humidity oversaturation is underestimated in this model (Krüger et al., 2022; Woiwode et al., 2020; Reutter et al., 2020). The latter has a significant impact on contrail formation (Gierens et al., 2020). Overall, earlier studies have shown that the EMAC model underestimates water vapour due to its cold bias (Brinkop et al., 2015; Jöckel et al., 2016). Our simulations reveal similarly that water vapour mixing ratios are consistently underestimated between 50 and 400 hPa, except for the tropopause region at around 200 hPa, where the EMAC model actually overestimates water vapour mixing ratios compared to ERA5. The difference between the ERA5 and EMAC increases towards the poles, with the two EMAC setups that include mean temperature nudging displaying an even larger deviation from ERA5. An explanation for this is that there is a rise in the wet bias in the extra-tropical area, which occurs due to water vapour diffusing horizontally from the tropical upper troposphere into the extra-tropical lowermost stratosphere. This phenomenon was also noted by Stenke et al. (2007). The overestimation is also evident in the comparison with observational data. The difference is significant for lower water vapour mixing ratios, which are typically present close to the tropopause. Aircraft often cross this region without flying directly along the tropopause. This results in very small sample sizes, which, in combination with strong lapse rate changes, make it a difficult region to evaluate.

Despite significant differences in water vapour mixing ratios, simulations with mean temperature nudging show smaller absolute differences in relative humidity over ice compared to ERA5 than those with standard nudging. However, both simulations display higher values than the reanalysis data. This indicates that the model temperature bias has a stronger effect on the relative humidity over ice than the model wet bias. Both biases directly impact predicted ice-supersaturated regions, which are crucial for the contrail life cycle. Simulations with standard nudging show larger areas for contrail formation compared to ERA5 across all relative humidity over ice thresholds. However, prior research has found that lower thresholds such as 90% (Dietmüller et al., 2023) or 93% (Hofer et al., 2024) achieve better agreement between observations and the ERA5 model results. This is consistent with the present study, where we found the highest correlation between the potential contrail coverage areas calculated online during the simulations using the Schmidt-Appleman criteria and the ISSR areas derived offline with a $RH_{ice}$ threshold of less than 96%. The order of magnitude of the identified area on the case study day is consistent with the ISSR occurrence results reported by (Petzold et al., 2020) for 250 hPa and 300 hPa, where they observed ISSR frequencies of occurrence of 19% and 25%, respectively. While the agreement between ERA5 and the various EMAC setups w.r.t. the potential contrail coverage area is within 10%, the difference to observations is greater. The point-by-point correlation between ML-CIRRUS measurement data and EMAC model setups regarding regions where conditions for persistent contrail formation



are possible indicates no correlation (low ETS score) between the model and observations when the model threshold is set to 90% while the observation threshold remains at 100%. This threshold pairing yields similar hit rates and false alarm rates

across the model setups. The model $RH_{ice}$ threshold significantly impacts both the hit rate and false alarm rate. Generally, higher thresholds tend to increase both metrics. However, maintaining a false alarm rate under 20% is preferable, as higher false alarm rates would result in more incorrect identification of ice-supersaturated regions, leading to unnecessary rerouting and an increased overall climate impact. The challenge is that achieving a low false alarm rate (<20%) typically results in a low hit rate (<50%), meaning many areas that actually allow contrail formation would not be detected. Despite this trade-off, it is

still preferable to prioritize a lower false alarm rate and accept a lower hit rate for identifying areas suitable for contrail formation, rather than having both rates high. It is important to note that the $RH_{ice}$ thresholds published by Dietmüller et al. (2023) and Hofer et al. (2024) pertain to ECMWF data, whereas the EMAC model might require a different threshold. Additionally, our sample size of 286 is relatively small compared to the study by Gierens et al. (2020). Nevertheless, similar Equitable Threat Scores and relationships between the measured and model data (indicating random correlations) are observed. In our case, a

higher ETS score (0.5) is only achieved by lowering both the model and observation thresholds below 90%. Li et al. (2023) previously found that contrail cirrus can also exist at relative humidity over ice values below 100%. However, since our study focuses solely on conditions for contrail formation, we use the 100% threshold for the observation data. Since the Schmidt-Appleman criterion, a thermodynamic theory that has been thoroughly tested and validated, is the basis for most calculations, any deviations mainly arise from inaccurate representations of key parameters for contrail formation, such as temperature or

humidity. Changes to the horizontal or vertical resolution of the model had only little effect on the outcome. The future studies need to evaluate the impact of the transition from the bi-linear interpolation to the nearest neighbor approach. Despite the fact that the 41-layer setup simulates the best agreement with observations, it still does not cover atmospheric processes in the stratosphere well, which could consequently have a minor influence on contrail prediction. However, this study reveals that this does not significantly impact the results, as all three model setups have the same model grid resolution of approximately

450 m vertically in the UTLS region, resulting in similar results. The cold and humidity biases present in all regions of the Northern Hemisphere that were shown in this study may not only affect contrail formation region predictions, but also have an impact on the water cycle, cloud formation, and the radiation budget.

Despite the presence of biases in the model, the areas predicted for contrail formation closely matched those observed in satellite images. This alignment may be attributed to the atmospheric conditions not being at the extreme edges, suggesting that

differences in atmospheric conditions between models and observations do not significantly impact the results in this case.

## 7    Summary and conclusions

This study aimed to evaluate distinct EMAC model setups for their capacity for investigating formation and radiative effects of contrails. Utilizing ERA5 reanalysis data, in-situ flight measurements, and satellite observations, we analyzed the effect of humidity and temperature biases in the model on contrail prediction. Our findings indicate that the vertical model resolution has

minimal impact on the model results. This suggests that the previously used EMAC model setup with 41 layers remains effec-



tive for future contrail studies, including the expansion of the geographic scope of contrail CCFs. Moreover, the improvements observed when increasing the horizontal grid resolution from T42 to T63 were too small to justify the disproportionate increase of required storage space and computational time. Despite the robustness of our results, some limitations exist. The model exhibits a temperature bias of up to 5 K, which can affect contrail prediction near the temperature threshold. Additionally, our study only analyzed data from the spring season of 2014, limiting the observation of long-term trends. Future research should aim to reduce the model bias, potentially by transitioning from the ECHAM to the ICON core model. It is also crucial to tune the mean temperature nudging (MTN) model setup towards a realistic radiation balance, as this technique strongly influences parameters such as humidity and the radiation budget. Failure to do so may render the model unsuitable for calculating the climate impact of contrails. In conclusion, this study enhances the understanding of contrail prediction limitations in current global climate models and provides an estimation of the accuracy and uncertainty when using temperature and humidity data in specified EMAC setups. The significant influence of temperature and humidity biases on contrail formation underscores the importance of considering model uncertainties when using climate-sensitive areas for contrail avoidance. Extending the existing Climate Change Functions to different seasons and regions is feasible and recommended for future studies.

*Code availability.* The Modular Earth Submodel System (MESSy, doi:10.5281/zenodo.8360186) is continuously further developed and applied by a consortium of institutions. The usage of MESSy and access to the source code is licensed to all affiliates of institutions which are members of the MESSy Consortium. Institutions can become a member of the MESSy Consortium by signing the MESSy Memorandum of Understanding. More information can be found on the MESSy Consortium Website (http://www.messy-interface.org).

*Data availability.* The ERA5 Data are available on the German Climate Computing Center DKRZ and the Copernicus Climate Data Store (https://climate.copernicus.eu/). The data of the ML-CIRRUS flight experiment was taken from the HALO Database (https://halo-db.pa.op.dlr.de/). Additional scripts and data are available upon request from the corresponding author. MSG/SEVIRI data can be obtained from EUMETSAT.

*Author contributions.* PP performed the EMAC model simulations advised by PJ; LB contributed the satellite images; AG and MK contributed the aircraft measurement data; SM, VG, PJ and CF were involved in the designed the study as well as in the discussion of the results and supported the writing of the paper. The paper was written by PP, all authors discussed and commented on the manuscript.

*Competing interests.* The authors declare that they have no conflict of interest. At least one of the (co-)authors is a member of the editorial board of ACP.



*Acknowledgements.* The individual authors of this study were supported by the ClimOP consortium within the framework of the European Union's Horizon 2020 research and innovation program, under grant agreement No. 875503. This study is also supported by the project D-KULT, Demonstrator Klimafreundliche Luftfahrt (Förderkennzeichen 20M2111A), within the Luftfahrtforschungsprogramm LuFo VI of the German Bundesministerium für Wirtschaft und Klimaschutz. We utilized high-performance computing resources from the German CARA Cluster in Dresden and the Deutsches Klima Rechenzentrum (DKRZ) Cluster in Hamburg. In preparing this manuscript, we used ChatGPT, an AI language model developed by OpenAI, to assist with text improvement and refinement.




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

**Appendix A:  Relative humidity and water vapour mixing ratio**

The relative humidity with respect to ice $RH_{ice}$ is a model parameter required to identify areas where contrails can form. This value relies on the specific humidity (SH), temperature (T), and pressure (P), and can be expressed as the ratio of the water vapour pressure (VP) to the saturation pressure over ice $SP_{ice}$ (see Equation A1):

$$RH_{ice} = 100 \cdot \frac{VP}{SP_{ice}} \tag{A1}$$

With the ratio of molar mass of water to molar mass of dry air ($\varepsilon$) being a constant of 0.622, the water vapour pressure is solely dependent on specific humidity and pressure, and is directly proportional to these variables (Equation A2), while the saturation pressure over ice can be estimated by the temperature (Equation A3, based on Murphy and Koop (2005)) - with the temperature being inversely proportional to the relative humidity.

$$VP = \frac{P \cdot \frac{SH}{1-SH}}{\varepsilon + \frac{SH}{1-SH}} \tag{A2}$$

$$SP_{ice} = \exp(9.550426 - \frac{5723.265}{T} + 3.53068 \cdot \log(T) - 0.00728332 \cdot T) \tag{A3}$$

This can be demonstrated in a simple example: When applying typical atmospheric values (temperature = 215 K, specific humidity = $3.5*10^{-6}$ kg/kg, pressure = 250 hPa) to Equation A1, the resulting relative humidity is approximately 101%. If the temperature is reduced by 5 K to 210 K (2.5%), while all other parameters are kept constant, the relative humidity over 810    ice rises to 200%. Alternatively, if the specific humidity doubles to $7*10^{-6}$ kg/kg, the relative humidity once again increases beyond 200%. Summarized, a decrease of temperature or increase of specific humidity will result in an increased relative humidity.

The water vapour mixing ratio (MR) is calculated from the specific humidity (SH) values in the model, using Equation (1); $\epsilon$ is the relation of molar mass of water to the molar mass of dry air (0.622):

$$MR = \frac{1}{\varepsilon} \cdot \frac{SH}{1-SH} \tag{A4}$$

The calculation is necessary for the comparison with humidity data from the ML-CIRRUS campaign, given as water vapour mixing ratio.