# Peer review of "Influence of Temperature and Humidity on Contrail Formation Regions in EMAC: A Spring Case Study"

_EGUsphere, 2024_

## Author Comment (AC1)

**Response to Reviewer Comments**

We thank the reviewers for their careful comments, which improved the quality of the manuscript. Below, the reviewer's comments are repeated in the *italic* text. Our response follows in normal letters. Blue text is used to cite from the revised manuscript. When page and line numbers are specified, they refer to the clean version of the revised manuscript.

**Reviewer 1 (RC1)**

*General Comments :*

*This study evaluates the EMAC model's ability to identify contrails formation regions. This is beneficial to setup operational mitigation options that avoid the climate sensitive regions. However, to some extent, the experiment result analysis is redundant and does not focus on the main conclusion. Some analyses can be removed. Furthermore, the overall presentation (including Figures and Tables) should be improved (see following Specific Comments). More importantly, merely providing a direct introduction to the simulation results can not meet the requirements of this journal. The current conclusions (e.g., a certain model setup shows the best agreement with observations) are only suitable to the EMAC mode version in this study. Deep analyses (e.g., physical mechanisms) are needed. The valuable conclusions must have generalization ability (e.g., mechanism analysis).*

We appreciate the reviewer's suggestion to improve the analysis by focusing more on physical mechanisms rather than presenting redundant results. In response, we have carefully revised the manuscript to streamline the discussion and remove repetitive sections, ensuring that the key findings are clearly presented and directly linked to our main conclusions. By restructuring parts of the results section, we now emphasize the role of temperature and humidity biases in contrail formation while avoiding unnecessary overlap in the discussion.

The reviewer also highlighted the need for a deeper physical analysis to enhance the generalization of our findings. We acknowledge this point and have expanded the discussion of the underlying physical mechanisms driving the observed biases. Specifically, we now provide a more detailed explanation of how the temperature-dependent saturation vapor pressure influences ISSR formation and contrail persistence. This additional analysis strengthens the connection between the model's biases and their impact on contrail formation, ensuring that our findings are not only specific to EMAC but also relevant to other GCMs and NWP models.

Regarding the concern that our conclusions might be overly specific to the EMAC version used in this study, we have revised the manuscript to clarify how our results contribute to broader contrail modeling efforts. We now explicitly discuss how similar temperature and humidity biases have been observed in other atmospheric models, reinforcing the applicability of our findings beyond this specific setup. Additionally, we highlight how the insights gained from this study can be used to refine future contrail avoidance strategies and improve model-based assessments of climate-sensitive regions.

The reviewer also suggested improving the presentation of figures and tables. Therefore, several figures and tables have been revised for clarity and improved readability.

Overall, we believe these revisions significantly enhance the manuscript by making the analysis more concise, physically grounded, and broadly applicable.

*Specific Comments:*

1. *Title: "EMAC" is not a well-known abbreviation. What is "EMAC" short for? This should be pointed out in the abstract.*
   *Abstract: It is necessary to point out which GCM was used in this study.*

   - We have revised the title to point out that the GCM "EMAC" was used in this study. In addition, we added an explanation about abbreviation EMAC in the introduction.
     [Title]: Influence of Temperature and Humidity on Contrail Formation Regions in the General Circulation Model EMAC: A Spring Case Study
     [Line 4-6]: This study evaluates key atmospheric parameters influencing contrail formation,  (temperature, humidity) using different setups of the ECHAM/MESSy Atmospheric Chemistry (EMAC) model. EMAC is a general circulation model  tested here with various vertical resolutions and two nudging methods for 7 specified dynamics setups.

2. *Abstract: "using various model setups of a general circulation model (GCM) with different vertical resolutions".  Since different resolutions are mentioned, the impact of resolution should be introduced.*

   - A higher vertical model resolution better represents the vertical structure around the tropopause, where large gradients in water vapour occur. In addition to cloud microphysics, this is essential for the simulation of contrails. We have added a line to the explain this.
     [Line 6-7]: A higher vertical resolution was tested to better capture the steep vertical gradients in water vapour near the tropopause, a factor crucial for accurate contrail prediction

3. *Abstract: "Accepting a false alarm rate of 16% results in a hit rate of about 40% (RHice threshold 99%), while aiming for an 80% hit rate increases the false alarm rate to at least 35% (RHice threshold 91-94%)" is very difficult to understand before reading the whole paper.*

   - Thank you for noting that the original description of false alarm and hit rates was difficult to interpret in the abstract. To address your concern, we have revised the abstract to present the key takeaway—namely, the trade-off between hit rates and false alarms—without going into detailed percentages that may be confusing without full context. The specific values and thresholds are now discussed in detail within the main body of the paper, where they can be properly explained. This adjustment should make the abstract clearer and more accessible to readers new to the subject.
     [Line 10-12]: ~~Exploring relative humidity over ice (RHice) threshold values for identifying ice supersaturation regions provides insights into the risks of false alarms for contrail formation, together with information on hit rates. Accepting a false alarm rate of 16% results in a hit rate of about 40% (RHice threshold 99%), while aiming for an 80% hit rate increases the false alarm rate to at least 35% (RHice threshold 91-94%.~~ Point-by-point comparisons along flown aircraft trajectories confirm similar biases, and sensitivity experiments with different relative humidity over ice thresholds illustrate the trade-offs between achieving a high hit rate and minimizing false alarms in contrail detection.

4. *Abstract: Right now, the conclusions from comparing simulation results with "reanalysis data for March and April 2014", "along flown trajectories with measurement aircraft data" and "a comprehensive one-day case" are separate. I think it is better to provide comprehensive conclusions rather than separate conclusions.*

- We restructured the abstract and made a more comprehensive conclusion.
  [Line 7-16]: Comparisons with reanalysis data for March and April 2014  reveal a systematic cold bias  (approximately 3–5 K in mid-latitudes) that persists unless mean temperature nudging is applied. In the upper-troposphere/ lower-stratosphere region, nudged simulations exhibit a wet bias, while lower altitudes show a dry bias , both affecting contrail formation estimates. Point-by-point comparisons along flown aircraft trajectories confirm similar biases, and sensitivity experiments with different relative humidity over ice  thresholds illustrate the trade-offs between achieving a high hit rate and minimizing false alarms in contrail detection. Integrating these insights, comprehensive one-day case study – incorporating aircraft based observations and satellite data – demonstrates that EMAC's predicted regions of potential contrail coverage  align well with observed contrail formation. Taken together, these results indicate that, despite existing temperature and humidity biases, EMAC provides valuable insights into contrail formation processes under various atmospheric conditions.

5. *Abstract: "confirms contrail detection in regions identified as potential contrail coverage areas by the GCM" is difficult to understand.*

- Thank you for bringing this to our attention. We have clarified the language in the abstract to avoid confusion about the relationship between the modeled and observed contrail regions.
  [Line 16-17]: Integrating these insights, comprehensive one-day case study – incorporating aircraft based observations and satellite data – demonstrates that EMAC's predicted regions of potential contrail coverage  align well with observed contrail formation.

6. *Abstract: At the last of abstract, it is better to give one useful suggestion based on the above evaluations.*

- Thank you for raising this point. We added suggestions at the end of the abstract.
  [Line 16-19]: Addressing these biases, for instance by refining temperature and humidity representation, could significantly enhance contrail prediction accuracy. Such improvements would not only strengthen confidence in contrail avoidance strategies but also support the development of more effective climate-optimized flight routing, ultimately helping to mitigate aviation's overall climate effect

7. *Line 24~25: Is only the EMAC model utilised to calculate climate change functions (CCFs)?*

- Just recently pycontrails were used for a similar approach. In the discussion section, the authors highlighted the similarities and differences of the approaches. We have added a short sentence to indicate this in the text.
  [Line 28-30]: Originally, the ECHAM/MESSy Atmospheric Chemistry (EMAC) model was utilized to calculate climate change functions (CCFs) (Grewe et al., 2014; Frömming et al., 2021). More recently, Engberg et al. (2024) employed a comparable methodology using pycontrails 0.51.0, enabling direct comparisons between the two approaches as discussed in their study.

8. *Line 30: Why must "a consistent modelling framework"?*

- In the revised manuscript, we now briefly explain why a consistent modelling framework is necessary, drawing on the reasoning provided by Grewe et al. (2017). Specifically, such a framework ensures that the climate responses derived from algorithmic climate change functions (aCCFs) are physically and methodologically coherent, making it possible to fairly compare different scenarios and emission types. This clarification has been added to the relevant section.
  [Line 37-44]: As discussed by Grewe et al. (2017), the use of CCFs in daily operations requires the implementation of fast algorithms in daily weather forecast systems, and their corresponding mitigation potential  must be evaluated in a consistent modelling framework.  Such a framework ensures that the derived climate responses to aviation emissions are physically and methodologically coherent, allowing a fair comparison across different scenarios and emission types. Achieving this requires an integrated setup that includes the algorithmic CCFs as well as an air traffic optimizer integrated into a general circulation model. So far, this approach has only been applied to NOx emission effects, and a validation exercise has been carried out to test the impact of NOx emissions on ozone (Yin et al., 2023; Rao et al., 2022).

9. *Line 31: What is the reason for "validation opportunities are limited"? Please give background introduction.*

- Apologies, we meant possibilities. The day-by-day calculation of CCFs is too time-consuming to allow an integration into daily weather forecast. Therefore, algorithmic versions (aCCF) were developed. A validation of those would imply a simulation with using aCCFs to drive an aircraft trajectory optimizer within a GCM simulation and simulate online the contrail formation to proof that the use of aCCFs actually reduces the contrail RF. Currently such a model version is available for NOx-emission effects, only and lacking for contrails. We have revised the text accordingly.
  [Line 37-44]: As discussed by Grewe et al. (2017), the use of CCFs in daily operations requires the implementation of fast algorithms in daily weather forecast systems, and their corresponding mitigation potential  must be evaluated in a consistent modelling framework.  Such a framework ensures that the derived climate responses to aviation emissions are physically and methodologically coherent, allowing a fair comparison across different scenarios and emission types. Achieving this requires an integrated setup that includes the algorithmic CCFs as well as an air traffic optimizer integrated into a general circulation model. So far, this approach has only been applied to NOx emission effects, and a validation exercise has been carried out to test the impact of NOx emissions on ozone (Yin et al., 2023; Rao et al., 2022).

10. *Line 45~48: These indicate that "temperature bias" is induced by "humidity bias (numerical diffusion of water vapour)". Am I right?*

- Yes, according to Stenke et al. (2007) and Charlesworth et al. (2023), the temperature bias in the upper troposphere/lower stratosphere (UTLS) is indeed likely linked to the numerical diffusion of water vapour across the tropopause. However, these studies also acknowledge that the atmospheric system is complex and that other processes influencing the hydrological cycle and atmospheric dynamics cannot be entirely ruled out. We will add this clarification in the revised manuscript to emphasize that while numerical diffusion of water vapour is a probable cause, additional factors may contribute to the observed temperature bias.
  [Line 56 – 60]: The EMAC model employed  here exhibits a temperature bias in the upper troposphere/lower stratosphere (UTLS) region (Stenke et al., 2007; Jöckel et al., 2016) , likely due to numerical diffusion of water vapour across the tropopause (Stenke et al., 2007) , a known issue in climate models (Charlesworth et al., 2023).  Nevertheless, because of the complex coupling between atmospheric dynamics and the hydrological cycle, other processes cannot be completely ruled out.

11. *Line 49: Since "temperature bias" is induced by "humidity bias", why nudge "temperature" rather than "humidity"?*

- We acknowledge that nudging temperature rather than humidity may appear counterintuitive. However, temperature is a more stable and robust variable to constrain at synoptic scales, especially when nudging is applied in spectral space. In contrast, water vapour exhibits larger spatial and temporal variability, making direct nudging less feasible and potentially destabilizing. Consequently, we follow a common practice in climate modeling by nudging temperature towards reanalysis data, while allowing the model's internal physics to handle humidity distributions at smaller scales. We have clarified this reasoning in the revised manuscript.
  [Line 60-63]: In the setups considered here, we nudge temperature rather than humidity. Temperature is more stable and less variable in time and space, making it more suitable for spectral nudging towards reanalysis data at synoptic scales. In contrast, the higher spatial and temporal variability of water vapour makes direct humidity nudging both infeasible and potentially destabilizing.

12. *Line 53~56: It is better to point out the purpose of these comparisons and evaluations.*

- Thank you for the comment. We have revised the text accordingly.
  [Line 67-72]: Second, we validate the model's ability to correctly identify regions where contrails are likely to form. We compare EMAC simulation results with atmospheric measurements, by analysing temperature and humidity data sampled along the trajectories of the High Altitude and Long Range (HALO) research aircraft during the ML-CIRRUS campaign. This comparison aims to assess how accurately the model reproduces observed conditions encountered in real flight scenarios. Finally, we evaluate satellite images, and compare  them with model predicted contrail-forming areas obtained from our  specified dynamics simulation.

13. *Line 118: Is potential vorticity (PV) usually used for seperating tropospheric and stratospheric? Why not separate tropospheric and stratospheric based on their definitions?*

- PV is commonly used for this purpose because it provides a dynamic representation of the boundary, which can be more consistent and reliable in certain contexts compared to static definitions based on altitude or temperature alone. The value of PV = 2.5 PVU (potential vorticity units) is a widely accepted threshold for defining the tropopause, particularly in mid-latitudes, and this method is supported in the literature. In the model the tropopause is a purely diagnostic quantity. We use the 2.5 PVU iso-line pole wards of 30°N/S and the WMO definition based on temperature lapse rate equatorward of 30°N/S. Thus, out choice is consistent with the model diagnosed tropopause in the extra tropics and polar regions.
  [Line 142-146]:  After sampling, we separate the data into tropospheric (P V ≤ 2.5) and stratospheric (P V > 2.5)  regimes based on the model's potential vorticity (PV).  PV is a commonly used metric for distinguishing between the troposphere and stratosphere, as it provides a dynamic and physically meaningful representation of the tropopause, particularly in mid-latitude regions. The threshold value of PV = 2.5  PVU was selected according to Zängl and Wirth (2002).

14. *Line 143~144: "This fraction is calculated as the difference between the potential coverage of contrails and cirrus clouds combined and the coverage of natural cirrus clouds alone." is difficult to understand. I still do not know how to calculate the fraction of contrail cirrus coverage. What is the difference between "the fraction of contrail cirrus coverage" and "the fraction of contrail cirrus coverage"? How to consider the sub-grid-scale variability in RHice?*

- The "fraction of contrail cirrus coverage" is represented as the "potential contrail coverage" (PotCov). PotCov is calculated as the difference between (i) the maximum possible combined coverage of contrails and natural cirrus and (ii) the coverage of natural cirrus alone. This isolates the potential contribution of contrails, determined by the Schmidt–Appleman Criterion (SAC), which uses local temperature and humidity thresholds.
  Sub-grid-scale variability can lead to localized RHi values above 100%, even if the grid mean is lower. Following Dietmüller et al. (2023), we use an RHi threshold lower than 100% for ERA5 data, which better matches observations (e.g., MOZAIC, Petzold et al., 2020). We have clarified this in the revised manuscript.
  [Line 159-170]: ~~Alternatively, the atmospheric ability to form persistent contrails and contrail cirrus in EMAC can be calculated at each time step according to Burkhardt et al. (2008) and Burkhardt and Kärcher (2009). Here, the fraction of a grid box available for contrail coverage is determined by the Schmidt-Appleman criterion, incorporating local temperature and humidity conditions conducive to contrail formation. This fraction is calculated as the difference between the potential coverage of contrails and cirrus clouds combined and the coverage of natural cirrus clouds alone.~~ One approach uses the parametrizations developed by Burkhardt et al. (2008) and Burkhardt and Kärcher (2009), as implemented in the EMAC CONTRAIL sub-model, to estimate the potential contrail coverage (PotCov). PotCov represents the fraction of a grid box that can be maximally covered by contrails under given large-scale temperature and humidity conditions. It is computed as the difference between (i) the maximum possible combined coverage of natural cirrus clouds and contrails, and (ii) the coverage of natural cirrus clouds alone. Both coverage values depend on critical temperature and humidity thresholds with respect to ice, which are adapted for large-scale model conditions. These thresholds also incorporate the Schmidt–Appleman Criterion (SAC; Schumann, 1996), which is derived from fundamental physical principles of mass, momentum, and energy conservation. In essence, the SAC determines whether hot exhaust gases mixing with cold ambient air will become supersaturated with respect to supercooled liquid water, enabling ice particle nucleation. This is often visualized using a 'mixing line' in the temperature–humidity space, which tracks how temperature and humidity evolve as exhaust and ambient air combine. Besides ambient temperature and humidity, the SAC accounts for aircraft-related parameters such as propulsion efficiency and fuel combustion heat.

15. *Line 178: What is the difference in the definition between cirrus and contrail cirrus? How to separate them?*

- Thank you for pointing out the need for clarification regarding the distinction between naturally occurring cirrus and contrail cirrus. In fact, there is currently no universally accepted microphysical definition that can reliably separate contrail-induced cirrus from natural cirrus clouds, especially once contrails have aged and transitioned into contrail cirrus. Young contrails can be identified by their high concentrations of very small ice crystals (less than about 10 µm in diameter), but as they evolve, these small particles gradually disappear and the microphysical properties of contrail cirrus become more similar to those of natural cirrus. In the ML-Cirrus study, the flight planning process incorporated contrail cirrus forecasts from the CoCiP model (Contrail Cirrus Prediction), which allowed us to select flight regions where contrail cirrus were likely to be encountered. This targeted approach enabled us to measure and compare both natural cirrus and contrail cirrus in situ. While subtle differences remain in the size distributions—particularly in the smallest ice particle size range—these differences would be difficult to attribute unambiguously to contrail origin without prior knowledge gained through predictive modeling and flight planning. We have clarified this in the revised manuscript.
  [Line 216-221]: Unlike young contrails, which can be distinguished from natural cirrus by their exceptionally high concentrations of very small ice crystals (diameters <10 µm), aged contrails (i.e., contrail cirrus) become micro physically similar to natural cirrus (Krämer et al., 2020). Consequently, identifying contrail cirrus solely by their properties is challenging. For the ML-Cirrus study, contrail cirrus forecasts from the CoCiP model Schumann (2012) guided the flight planning process, allowing the targeted sampling of regions where contrail cirrus were likely to occur and enabling a comparative analysis of both cloud types.

16. *Line 197~203: Here, only the treatment of the difference in time interval is introduced. How about the spatial difference between ML-CIRRUS and model results?*

- We have now clarified both the temporal and spatial handling of the ML-CIRRUS observations in the revised manuscript. In addition to describing the time averaging from 1-second to 10-minute intervals, we have included details on how the EMAC submodel S4D (Joeckel et al. 2010) is used to spatially interpolate the model data to the exact geographical locations of the averaged observational data points. This approach ensures that the model output and observations are aligned not only in time but also in space, thereby minimizing representativeness errors. We believe these revisions address your concerns and provide a clearer understanding of how we achieve a consistent comparison between model results and measurements.
  [Line 245-249]: For spatial alignment, we used the averaged positions derived from the ML-CIRRUS flight track data as input to the EMAC submodel S4D (Jöckel et al., 2010). The S4D submodel interpolates the three-dimensional model fields to the exact geographical location and time corresponding to each 10-minute averaged observational data point. This procedure ensures that the model data are sampled at the correct geolocation thereby minimizing spatial representativeness errors.

17. *Line 206: Why only 26 March 2014? The ML-CIRRUS campaign from March 10 to April 16 2014.*

- Yes, the ML-CIRRUS campaign lasted for several weeks, but only two days with contrails and contrail cirrus were probed Voigt et al. (2017), on 26.3.2014 (this case study) and on 10.4.2014. However, on the second day the contrail outbreak was less pronounced in the satellite observations and in particular it was probed by the HALO only in the afternoon when most contrails were already dissipating or had dissipated. In contrast, the selected day for our study has been investigated in detail in wang et al (2023) using in situ cloud probes, airborne lidar observations and MSG/SEVIRI showing that many contrails were present in the area West of Irland. Thus, this case during ML-CIRRUS is much better suited to the scope of this study than the other one and this is why we focus on it. A better clarification for the selection of this day has been added to the manuscript.
  [Line 254-258]: The SEVIRI instrument aboard the geostationary Meteosat Second Generation (MSG) satellite is used to observe contrails in the region probed by the HALO aircraft during ML-CIRRUS on 26 March 2014. This is the most prominent contrail outbreak encountered during the ML-CIRRUS campaign and has been investigated using in situ cloud probes, airborne lidar observations and MSG/SEVIRI data in Wang et al. (2023). Thus, this day is very well suited for a comparison with the climate model data investigated in this study.

18. *Line 235~248: Compared to STN setups, the temperature bias from MTN setup is almost negligible. In other words, the difference between STN and MTN is obvious. The reason for this difference must be clearly illustrated.*

- We have clarified the underlying reason for the reduced temperature bias in the MTN setups. In the revised manuscript, we explain that while STN configurations only nudge selected meteorological variables without enforcing an accurate global mean temperature, the MTN setup includes an additional constraint on the global mean temperature (wave-zero mode). This additional constraint prevents the model from drifting toward a systematic global temperature offset. Consequently, the MTN configuration maintains a closer alignment with the reanalysis data, resulting in an almost negligible temperature bias.
  [Line 287-302]: Figure 2 illustrates the area weighted mean temperature across six distinct EMAC model setups (see Table 1) for the extended NAFC between 50 hPa and 400 hPa in comparison with ERA5 for March and April 2014. Notably, our comparison reveals a cold bias in EMAC simulations with standard temperature nudging compared to ERA5 reanalysis data for the analysed region. The identified bias for the area-weighted average mean temperature data ranges from 3 K to 5 K between 400 hPa and 50 hPa, with the most significant discrepancy around 200 hPa. Increasing the spectral (horizontal) resolution from T42 to T63 leads to a minimal reduction of the bias, by up to 0.2 K, while changes in vertical resolution have negligible effect. The STN model setup with 41 vertical layers exhibits the strongest deviation from ERA5, being 0.1 K colder than other STN setups up to 100 hPa. The analysis of 20∘ latitude bands for the NAFC (Figure 4, left) indicates a consistent altitude-dependent temperature bias across all bands. However, the application of mean temperature nudging (T42L31MTN, T42L41MTN), as explained in Section 2.1, significantly reduces, as to be expected, the cold bias to less than 0.1 K across all analysed pressure levels and bands. Since the STN setups constrain only selected meteorological variables (e.g., vorticity, divergence, temperature, and surface pressure) to large-scale reanalysis patterns without enforcing an accurate global mean temperature, a systematic global temperature offset can still develop. In contrast, the MTN setup imposes an additional constraint on the global mean temperature, preventing such a systematic bias from emerging. As a result, MTN keeps the model closely aligned with the reanalysis, making the temperature bias almost negligible. In contrast, the MTN setup imposes an additional constraint on the global mean temperature, preventing such a systematic bias from emerging. As a result, MTN keeps the model closely

aligned with the reanalysis, making the temperature bias almost negligible. In the MTN setup, the bias is only present above 80 hPa, in a region where no nudging is applied.

19. *Figure 3: It is necessary to show the probability density from ERA5 data (i.e., benchmark). Furthermore, I think it is better to use line chart rather than bar chart (line chart is clearer than bar chart, especially for the difference among different EMAC model setups).*

- Thank you for your comment. Figure 3 is designed to highlight the differences between the various EMAC model setups relative to ERA5. To avoid overcrowding this plot, we present the probability density of ERA5 along with the other EMAC datasets separately in the supplement (Figures 15–19). Regarding the use of bar charts, we believe that this visualization effectively illustrates the differences among the EMAC model setups, making it easier to compare their variability.
  [Line 319-323]: When looking at PDFs of the ERA5 and EMAC model setups separately (Figure S15, S16, and S17), a shift toward higher humidity values in EMAC setups compared to ERA5 is noticeable at all pressure levels. The altitude-dependent cold bias can partly explain the disparity in relative humidity between ERA5 and EMAC in the STN simulations, as the saturation vapour pressure over ice depends solely on temperature (A3)

[Figure]

20. *Line 250〜257: For me, it is very dificult to see these conclusions from Figure 3.*

- Thanks for the comment. We adapted the text to make it clearer.
  [Line 308-313]: The MTN simulations, on average, exhibit closer agreement with ERA5 than the STN simulations, with noticeable differences across pressure levels.  At: 200 hPa,  MTN simulations differ typically by 5-12%, while  STN differ typically by 12-17: %. At 250 hPa and 300 hPa, differences between all STN model setups are further reduced, while the difference between EMAC and ERA5 remains consistent on average.  Analysis shows that the vertical resolution, does not play a central role, but increasing the spectral (horizontal) resolution to T63  diminishes the difference to ERA5 across all  investigated: pressure levels.

21. *Line 271: "For MTN simulations, a consistent wet bias is observed at higher altitudes (150 hPa to 250 hPa)". What is the reason?*

- The precise cause of this wet bias at higher altitudes is not fully understood and would require additional targeted analyses and sensitivity simulations, which lie beyond the scope of the present study. We can only speculate based on previous findings by Joeckel el al. (2016) that the wave-zero temperature nudging introduces a substantial external forcing to the model. This strong forcing may pull the system away from its internally balanced state, prompting compensatory adjustments in other dependent variables. Moreover, it is important to note that the reanalysis data used for nudging are not necessarily fully consistent with our model's own physics and parameterizations. We discuss this now in the manuscript.
  [Line 577-580]: While the exact cause of the observed wet bias for the MTN simulations at higher altitudes remains uncertain, it may be related to the wave-zero temperature nudging forcing the model away from its internal equilibrium, prompting compensatory adjustments in humidity fields. Further dedicated analyses or sensitivity simulations, beyond the scope of this study, would be needed to substantiate this explanation.

22. *Line 309: "the area-weighted mean potential contrail coverage (AWM PotCov) for grid boxes with a potential contrail coverage above zero" is difficult to understand.*

- Thank you for pointing this out. We changed the text and made it clearer.
  [Line 373-376]: The influence of  vertical model resolution and nudging methods on the  PotCov approach is also visible in Table 2 for the NAFC. While  area-weighted mean  PotCov remains consistent across  models (0.36 ± 0.02 at 250 hPa, 0.35 ±0.04 at 300 hPa, 0.34 ± 0.03 at 350 hPa),  total coverage varies. At 250-300 hPa , simulations with  31-41levels show coverage

between 59-53%. At 350 hPa, both MTN setups  have lower coverage than the STN setups .

23. *Line 331~338: The ML-CIRRUS Campaign provides in-situ measurements (i.e., a trajectory line), while the model results along trajectories denote the model grid-mean variables (i.e., a large area). How to consider the difference between "in-situ measurements" and "model grid-mean"?*

- We appreciate the reviewer's comment regarding the difference between in-situ measurements and model grid-mean values. In the revised manuscript, we have clarified how these differences are addressed. Specifically, we have highlighted that ML-CIRRUS measurements provide localized, high-frequency data along the aircraft's trajectory, whereas the model represents conditions averaged over larger spatial domains. To reduce this scale mismatch, we now explicitly state that we average the ML-CIRRUS data over 10-minute intervals, matching the temporal resolution of the model output. This smoothing approach helps mitigate the impact of high-frequency variability in the observational data and allows for a more meaningful comparison with the model's grid-mean conditions. We believe this added clarification and methodological detail address the reviewer's concerns.
  [Line 393-404]: In general, the climate impacts of non-CO2 effects are complex and require detailed knowledge about key atmospheric parameters. Nevertheless, numerous difficulties may arise when modelling parameters such as temperature or humidity using a general circulation model (Jöckel et al., 2016). Consequently, it is advisable to utilize atmospheric measurements to assess model performance in particular regions. Hence, we compared temperature and humidity data on aircraft trajectories obtained during the ML-CIRRUS campaign with the emulated trajectory data from different model setups. It is important to note that the ML-CIRRUS campaign provides in-situ measurements along specific aircraft trajectories, representing localized observations, whereas the model results are grid-box mean values, which represent average conditions over larger spatial areas. To partly account for these differences, we averaged the ML-CIRRUS data over 10-minute intervals, aligning with the model output frequency, thereby smoothing the high-frequency variability of the in-situ data. This approach allows a more consistent comparison, though some discrepancies may still arise due to the inherent spatial and temporal scale differences. From the measurement data, we derive potential contrail formation conditions (e.g., ice-supersaturation) that can be compared with EMAC model potential contrail areas and validated with satellite imagery.

24. *Line 340: How to calculate the correlation? Is the correlation based on time series or spatial distribution? Why not compare temperature data directly based on the trajectory map?*

- Thank you for your comment. The correlation was computed using point-by-point comparisons between the in-situ temperature measurements obtained during the ML-CIRRUS flights and the corresponding model output at the exact same geographical location and time. For each measurement point along the flight trajectory, we extracted the model temperature from the nearest spatial and temporal grid point. By doing so, we created a paired dataset of measured and modeled temperatures, and then calculated the correlation coefficient across all these pairs.

In other words, the correlation is not solely based on a time series or a spatial distribution taken independently, but rather on co-located in situ observations and model predictions. This approach ensures that the correlation reflects how well the model reproduces observed conditions at the specific times and places where measurements were taken. While comparing temperature data directly on a trajectory map might offer a visual assessment, the statistical correlation using matched data pairs provides a more quantitative measure of model performance.

[Line 406-410] All model setups exhibit a  high degree of agreement with the observed in situ temperature data from the ML-CIRRUS  flights (Fig. 5. ). To assess this agreement, we first applied spatial resampling by extracting EMAC model data along the aircraft trajectories, followed by temporal resampling of the high-resolution aircraft data to the model's time steps. The resulting time series were then compared, yielding Pearson correlation coefficients (r) between 0.98 and 0.99. This high correlation indicates an excellent correspondence between observed and modelled temperature.

25. *Line 349~351: "The differences between the MTN simulation results and the measurements can mainly be attributed to the resampling of the observation data and the uncertainties during the measurement process". I can not understand this reason. Please give a more detail explanation.*

- Thank you for highlighting the need for clarification. The differences between the MTN (mean temperature nudging) simulation results and the measurements stem from two primary factors: (1) data resampling, which can introduce smoothing when matching observational data points to the model output intervals, and (2) measurement uncertainties, such as sensor calibration and environmental conditions. These issues collectively reduce the precision of the observed humidity and temperature values and can lead to mismatches between model and observation data.

  [Line 419-424]: The differences between the MTN simulation results and the measurements can  be attributed to the resampling of observation data and uncertainties during the measurement process. Resampling aligns observational points with model output intervals, which can lead to data smoothing and loss of fine-scale information, especially in rapidly changing conditions. Additionally, measurement uncertainties—such as sensor calibration, environmental factors, and instrumental limitations—impact the precision of observed humidity and temperature values, further contributing to the discrepancies.

26. *Figure 5: the Figure S5a and S5b can be included in Figure5. Some other figures can also be improved in this way.*

- Thank you for your comment. Figure S5a and b are now part of Figure5.

[Figure]

27. *Line 355~365: The relations between Figure 6 and Figure 2 should be mentioned.*

- Thank you for pointing that out. The reduction of the temperature bias does also to reduce the water vapor mixing ratio difference between measurement and model data. We have added this in the manuscript.
  [Line 425-433]: The analysis of water vapour mixing ratios across various model setups and measurement data, illustrated in Figure 6, reveals  notable discrepancies between model results and observations. For high levels of humidity ranging between 100 and 800 ppmV, predominantly present in the upper troposphere, the results from the standard temperature nudging simulations persistently provide lower values than the HALO FISH and SHARC measurements (see Figure 6, left). This "dry bias" is strongly reduced in simulations with mean temperature nudging (see Figure 6, right). This could be directly connected to the temperature bias reduction found in Figure 2, as temperature influences humidity. In contrast, for low humidity values between 5 to 10 ppmV, which are located in the lower stratosphere and are not directly relevant for contrail formation, the model consistently overpredicts compared to the observations by up to six times across all model setups. This model "wet bias" appears with both nudging methods.

28. *Line 373: Table 3 should list comparisons based on the model results of the some other threshold (e.g., 85%,95%,and 100%).*

- Thank you for your comment. We have provided the link between Figure 7 and Table 3. Figure 7 presents Hit rates (HR) and FA (false alarm) for eleven different threshold values between 90% and 100%.
  [Table 3] The values HR and FA are illustrated in figure 7, together with different threshold between 90 and 100%.

29. *Line 379: "The Equitable Threat Score (ETS) characterizes the agreement between the data sets, considering hits, misses, and false alarms while adjusting for random chance" is difficult to understand.*

- Thank you for the suggestion. We have clarified the description of the Equitable Threat Score in the manuscript. The revised explanation now briefly defines the ETS, highlights that it accounts for hits, misses, and false alarms, and emphasizes that it is adjusted to remove the influence of random chance, making it a more equitable measure of agreement between the datasets.
  [Table 3] The Equitable Threat Score (ETS) is a commonly used verification metric that evaluates how well two datasets align by considering not only correct predictions (hits) but also incorrect ones (misses and false alarms). It adjusts for agreements that could occur by chance, providing a fairer measure of predictive skill (ref to Ebert (1996)).

30. *Line 404~406: It seems that the selected case (i.e.,26 March 2014) is not common case. In my opinion, it is better to choose a representative case.*

- Thank you for raising this point. We have revised our description in section 5; wording for describing the meteorological situation was a bit misleading, by using the term "intriguing", which we replace with the more adequate word "interesting"; and we suggest to mention the specific flight profile on that day, performing several vertical dives. Specifically, first, 26 March 2014 featured an interesting meteorological situation (synoptical situation in Europe), the specific region we investigated on that day was not significantly affected by the polar cold front outbreak. In fact, the tropopause height of around 200 hPa is quite typical for this latitude, making the local conditions representative. Second, the three flight dives conducted on this day provide a valuable opportunity to investigate differences between the troposphere and stratosphere under slightly varying atmospheric conditions. These dives demonstrate how changes in flight maneuvers influence the observed temperature and humidity profiles. For these reasons, we consider 26 March 2014 to be a suitable and informative case for our analysis, and have updated description in section 5 accordingly. Additional reasons for choosing this day are provided in the response to review question #17.
  [Line 480-484]: As a case study, we selected 26 March  2014. Although a polar cold front outbreak  occurred across Europe on this date, the region examined here remained largely unaffected by these unusual conditions, maintaining a typical tropopause height of around 200 hPa for this latitude.

Moreover, the three flight dives performed on this day allowed us to systematically explore variations between the troposphere and stratosphere and to assess how different dive strengths influenced the observed temperature and humidity profiles.

31. *Line 409: "the methodology of REACT4C" should be introduced.*

- Thank you for pointing out the need to clarify the methodology of REACT4C. After reviewing the relevance of the REACT4C approach in our manuscript, we decided to remove the reference entirely to maintain focus on the primary objectives of our study.
  [Line 490]

32. *Line 410: "Geopotential height anomalies at 250 hPa indicate minimal correlation with both, the North Atlantic oscillation index, and the East Atlantic index". What supports this conclusion?*

- Thank you for pointing this out. We have removed the statement regarding the correlation with the North Atlantic and East Atlantic indices. Without a more robust analysis to support this conclusion, it no longer appears in the revised manuscript.
  [Line 488-89]: The weather situation over the North Atlantic on the 26 March  2014 shows a relatively weak jet stream, which is limited to the western North Atlantic

33. *Line 423~424: The word "parameter" might lead to misunderstanding (e.g., the tunable parameters in model code).*

- *We understand that the word parameter is misleading here and changed the sentence in the manuscript accordingly.*
  *[Line 498-499]: A direct analysis along the trajectory provides insight into  variations in specific atmospheric variables that are not apparent in larger dataset statistics.*

34. *Line 440: "during the third dive, the observed RHice are lower and the model-predicted RHice values exceed the observational data in parts by roughly 20%. This suggests that the simulated gradient is not steep enough, as the flight is further away from the tropopause than in the dives before" is difficult to understand.*

- Thank you for pointing that out. We made it clearer.
  [Line 513-515] However, during the third dive, the  model-predicted $RH_{ice}$ values exceed the observational data  by roughly 20%.  Unlike the previous dives, the third dive occurs further from the tropopause , potentially impacting the accuracy of the model's representation of the relative humidity.

35. *Line 465: If possible, please show air traffic density in Figure 9. At least, show the airports and flights in Figure 9.*

- We appreciate the reviewer's suggestion to include air traffic density and specific airports/flights in Figure 9. We believe that showing incorporating information on all airports and other flights falls outside the scope of our analysis. Instead, we have added the trajectory of the ML-CIRRUS flight (shown in purple) to illustrate a relevant in-situ measurement in the area of interest. This addition highlights the contrail conditions measured on that day without overcomplicating the figure. We hope this revised figure provides sufficient context to interpret our results.

[Figure]

36. *Line 498: What is anti-ice (AI) correction?*

- We thank the reviewer for pointing out the need to clarify the term "anti-ice (AI) correction." We have revised the manuscript to include a detailed explanation of the AI correction in the context of aircraft temperature measurements. The added text provides background on how the anti-ice heating of the temperature probe affects the measurements and why this correction is necessary.
  [Line 586-595]: Recent findings have provided new insights into the anti-ice (AI) correction for aircraft temperature measurements (Giez et al., 2023), indicating a possible cold bias in the temperature data, which we want to discuss here. The static air temperature on aircraft is determined using a Total Air Temperature (TAT) probe, which measures the temperature of the air impacted by the aircraft's motion. This probe is housed within a sensor inlet (e.g., Collins Aerospace 102BX) that is actively heated to prevent ice formation, a process known as Anti-icing. Ice accumulation on the inlet can lead to erroneous temperature measurements under in-flight icing conditions. The heating affects the temperature readings inside the housing, necessitating a correction known as the anti-ice (AI) correction. While the manufacturer provides a parametrized correction for this effect, the AI correction for the HALO aircraft was determined individually through in-flight calibration (Bange et al., 2013). During the ML-CIRRUS campaign, no AI correction was applied to the measurement data, based on experimental evidence suggesting it was unnecessary.

37. *Line 553: "However, since our study focuses solely on conditions for contrail formation, we use the 100% threshold for the observation data." I can not understand the logical relationship.*

- Thank you for pointing out the need for clarification. The distinction here is between contrail formation and contrail persistence. While prior studies (e.g., Li et al., 2023) have shown that existing contrail cirrus can persist even at relative humidity over ice (RHi) values below 100%, initial contrail formation generally requires saturated or nearly saturated conditions (i.e., around 100% RHi). In our study, we are specifically focused on the conditions necessary for contrail formation rather than on their subsequent evolution. Therefore, we use a 100% RHi threshold for the observational data to ensure that we are examining the formation phase under realistic conditions. Any exploration of lower thresholds in our analysis was intended to test sensitivity and improve model verification scores but does not reflect the physical conditions strictly required for the onset of contrails.
  [Line 634-636]:  While contrail cirrus can persist below 100% RHice (Li et al., 2023), the initial formation phase typically requires conditions near saturation. Thus, our use of a 100% observational threshold is physically justified, and lower tested thresholds were intended only to assess model sensitivity.

38. *Line 564~565: "This alignment may be attributed to the atmospheric conditions not being at the extreme edges, suggesting that differences in atmospheric conditions between models and observations do not significantly impact the results in this case." is difficult to understand.*

- Thank you for highlighting the confusion in this sentence. We intended to convey that the atmospheric conditions on this particular day were not near the thresholds at which small differences in temperature or humidity would critically alter contrail formation. As a result, even though the model exhibits some biases, those biases did not significantly shift the areas where contrails could form. To clarify this, we have revised the manuscript.
  [Line 659-665]  Despite the model's biases, the predicted contrail formation areas closely match those observed in satellite imagery. From 03–05 UTC, we find close agreement in both the high-coverage contrail regions and the regions without contrails. We attribute this agreement to atmospheric conditions being sufficiently far from critical thresholds for contrail formation, so temperature and humidity biases do not significantly affect the ISSR evaluation. Later in the day, from 07–09 UTC, we find only partial agreement in the high-coverage contrail regions, likely due to atmospheric conditions being closer to the formation threshold. Under these conditions, any model bias affects the ISSR evaluation, and areas where contrails are observed are not consistently captured by the model.

39. *Discussion: This study evaluates the EMAC model's ability to identify climate-sensitive regions. However, the final demand is to enable their forecast within numerical weather prediction models. Can the global mean temperature nudging technique be used for model prediction simulation? Which conclusions from this study are still useful for numerical weather prediction models? It is necessary to discuss this issue.*

- While our study is primarily focused on evaluating EMAC, a climate chemistry model, we intend to discuss to what extent lessons learned regarding temperature and humidity biases are directly applicable to numerical weather prediction (NWP) models. We now explain in the discussion (and conclusion) the importance to assimilate realistic temperature and, if possible, also humidity data in the UTLS region. The global mean temperature nudging (MTN) technique, as applied here, is not typically a standard feature of NWP models, which rely heavily on data assimilation techniques to provide accurate initial conditions. However, the principle that systematic temperature and humidity biases must be addressed to improve the reliability of contrail predictions remains relevant. In NWP contexts, biases discovered in climate models can guide improvements in data assimilation schemes, bias correction methods, or parameterization adjustments. The insights from this study—such as the sensitivity of contrail formation to subtle changes in temperature and humidity, the limited benefit of increasing model resolution without tackling underlying biases, and the importance of carefully selected humidity thresholds—can inform the refinement of NWP models' initial conditions or their parameterizations for contrail forecasting.
  [Line 665-669]: Although the global mean temperature nudging used in EMAC may not directly translate to operational numerical weather prediction (NWP) systems, the overarching conclusions remain relevant. NWP models could employ improved data assimilation, bias correction strategies, and carefully chosen RH thresholds to enhance contrail forecasting. By doing so, future operational frameworks might better integrate contrail avoidance measures, ultimately contributing to climate-optimized flight operations.

40. *Line 568: This study did not evaluate the radiative effects of contrails.*

- Thank you for highlighting this discrepancy. We recognized that the radiative effects of contrails were not directly assessed in our analysis, so we have removed the mention of radiative effects in the revised manuscript. The focus now remains solely on the evaluation of contrail formation in the different EMAC model setups.
  [Line 656-656]  This study enhances the understanding of contrail prediction limitations in the global climate model EMAC and evaluates categorical statistical measures related to temperature and humidity biases in different EMAC setups.

41. *Line 576: "Future research should aim to reduce the model bias, potentially by transitioning from the ECHAM to the ICON core model". What supports this conclusion?*

- We recognize that transitioning from the ECHAM to the ICON core model extends beyond the scope of the present study since we did not directly compare ICON-based simulations or evaluate its potential for reducing the biases identified. Therefore, we have removed the reference to moving toward the ICON core to keep the discussion focused on the main findings of this work.
  [Line 669]

42. *Line 579~581: Can the simulation results from only one model (i.e., this study) support these conclusions (i.e., GCMs)?*

- Thank you for highlighting this point. We acknowledge that our study focuses on the EMAC model specifically and does not directly address all GCMs. To avoid overgeneralizing, we have revised the text to reflect that our conclusions primarily pertain to EMAC rather than GCMs in general. This clarification ensures that readers understand the scope and applicability of our findings.
[Line 685-686]: Consequently, **EMAC** reliably identifies potential contrail-forming regions, demonstrating its suitability for contrail studies.

43. *Line 581~583: "The significant influence of temperature and humidity biases on contrail formation underscores the importance of considering model uncertainties when using climate-sensitive areas for contrail avoidance. Extending the existing Climate Change Functions to different seasons and regions is feasible and recommended for future studies". I can not understand the logical relationship.*

- Thank you for pointing out the need for clearer explanation. We have deleted mentioning planned expansions of CCFs, nevertheless mentioning significance of our results for provision of aCCFs which rely on identification of climate-sensitive areas for contrail avoidance from NWP data.
[Line 692-696] Nevertheless, the temperature and humidity biases identified in this study contribute to forecast uncertainties in contrail modelling, which need to be addressed when predicting ISSRs in atmospheric models. These uncertainties form the basis for risk assessments when implementing contrail avoidance strategies or climate-optimized trajectories. By extending existing Climate Change Functions (CCFs) to different seasons and geographical regions, future studies can better quantify how such uncertainties vary over time and space

*Technical Corrections:*

44. *Line 29: "climate change functions" should be replaced by "CCFs".*

- Thank you for highlighting this. We have replaced 'climate change functions' with the abbreviation 'CCFs' in the mentioned line to maintain consistency throughout the manuscript.
[Line 37]: As discussed by Grewe et al. (2017), the use of  CCFs in daily operations requires the implementation of fast algorithms in daily weather forecast systems

45. *There are too many abbreviations in this study. Many of them are unnecessary.*

- Thank you for noting the excessive use of abbreviations. We have reviewed the manuscript and removed or spelled out several abbreviations that were not essential to the discussion. This should make the text more readable and reduce potential confusion for the reader.

46. *Line 419: "Figure 9" appeared earlier than "Figure 8".*

- Figure 8 appears now earlier than Figure 9.

**Reviewer 2 (RC2)**

*This study tests the ability for EMAC to accurately model contrail occurrence based on meteorological conditions. The evaluations are extensive and appear well-conducted despite mixed results. The purpose of this study is currently quite technical and I request the text be made clearer with a key result explained in more detail, though with some changes I expect the study will be suitable for ACP.*

**General Comments**

*The skill values (ETS) in Table 3 are very low. The presentation of results here seems appropriate, but surely there is more behind the low skill than the stated reason that this "might be due to the small number of data points" (Line 399). This lack of skill is also possibly suggestive of limits on the methods of this study. For instance, there may be improvement if the same test were applied with RH values from ERA5 rather than EMAC (zero bias by the criteria in Figs. 2-4), but presumably ERA5 also has some RH bias given a dearth of direct observations at these high altitudes. Could the authors please add to their explanation, and also please discuss future avenues for obtaining more accurate contrail estimation in such a comparison?*
*The study is presented according to a quite technical purpose (assessing two parameters of contrail occurrence in this specific model), but if there are more general insights to convey, this would bolster the study's suitability for this scientific journal. All the results here currently seem specific to use of the EMAC model. This prompts questions, e.g. are similar biases apparent in other models? Are there conclusions to take from this work that are not model-specific? I'm wondering, for instance, if the low skill scores mentioned above speak generally to a limited current ability to accurately model contrail occurrence?*

Thank you for highlighting the low ETS values and the potential limitations they suggest. In the revised manuscript, we have expanded our discussion to clarify that the low skill scores cannot be solely attributed to the small sample size. We now explain additional factors, including the inherent difficulty of comparing localized in-situ observations to grid-box averaged model values, the identified temperature and humidity biases, and the complexity of modeling contrail formation processes. We also acknowledge that even reference data sets like ERA5 may exhibit their own humidity biases, which complicates direct verification.

Furthermore, we have added suggestions for future improvements. These include refining $RH_{ice}$ thresholds for EMAC, applying bias correction techniques, incorporating more comprehensive observational data sources (e.g., additional flight campaigns or satellite measurements), and enhancing data assimilation methods and parameterizations. By taking these steps, we hope to increase the accuracy and reliability of contrail predictions in subsequent studies.

We acknowledge that many of our results, at first glance, appear specific to EMAC. In the revised manuscript, we have taken steps to highlight broader implications that extend beyond this particular modeling framework.

First, we discuss how the temperature and humidity biases identified in EMAC are also encountered in other state-of-the-art models and reanalysis datasets, as indicated by previous studies. These issues stem from common challenges in representing atmospheric processes, such as ice-supersaturation and cirrus formation, and are not limited to a single model architecture. By noting the prevalence of similar biases in ERA5 and referencing related findings in other global climate and weather models, we illustrate that the problem is systemic rather than isolated to EMAC.

Second, we emphasize that the low skill scores in matching observed contrail formation likely reflect more general limitations in current modeling capabilities. The complexity of small-scale humidity variability, sensitivity to microphysical thresholds, and dependence on parameterizations of aircraft engine characteristics are difficulties that any model would face. Consequently, our results suggest that more refined parameterizations, improved data assimilation, and targeted bias correction strategies could enhance contrail prediction in a wide range of models.

Finally, we connect our findings to the broader context of operational forecasting and climate-optimized aviation routing. By showing where and why discrepancies arise, we provide insights that could guide future model development efforts, irrespective of the chosen modeling system. We believe these broader lessons, now more explicitly stated, will help readers see the study's relevance for improving contrail modeling and forecasting in general.

**Specific Comments**

1. *Line 1: more of an 'uncertain' than "significant" impact of contrails, as some estimates indicate a negligible forcing but they are overall varied. C.f. Bier & Burkhard 2022, Bock & Burkhardt 2016 (0.056 Wm/2), Chen & Gettelman 2013 (0.013 W/m2), Rap et al 2010 (<0.01 W/m2)*

   *Bier, A., & Burkhardt, U. (2022). Impact of parametrizing microphysical processes in the jet and vortex phase on contrail cirrus properties and radiative forcing. Journal of Geophysical Research: Atmospheres, 127(23), e2022JD036677.*

   *Bock, L., & Burkhardt, U. (2016). Reassessing properties and radiative forcing of contrail cirrus using a climate model. Journal of Geophysical Research: Atmospheres, 121(16), 9717-9736.*

   *Chen, C. C., & Gettelman, A. (2013). Simulated radiative forcing from contrails and contrail cirrus. Atmospheric Chemistry and Physics, 13(24), 12525-12536.*

   *Rap, A., Forster, P. M., Jones, A., Boucher, O., Haywood, J. M., Bellouin, N., & De Leon, R. R. (2010). Parameterization of contrails in the UK Met Office climate model. Journal of Geophysical Research: Atmospheres, 115(D10).*

   - Thank you for your comment. We have revised the abstract to better reflect the uncertain nature of the radiative forcing from contrails. Specifically, we replaced "significant impact" with "impact" and we modified the phrasing to highlight that, despite this uncertainty, contrail avoidance is currently discussed as a valid strategy for mitigating aviation-related climate impacts.
   [Lines 1-3]: While carbon dioxide emissions from aviation often dominate  aviation-related climate change discussions, the  impact of non-$CO_2$ effects , particularly contrails and contrail-cirrus, *:* must not be overlooked . Despite varying estimates of their radiative forcing, avoiding contrails is currently discussed as a valid strategy for reducing these climate effects.

2. *Line 18: This would be read more cleanly if it just focused on the role of contrails, rather than as a component of "non-CO2 effects".*

- Right, this paper focusses on contrails. However, we think that it is worth mentioning that contrails are a part of the non-CO2 effects. In the revised version we highlight the variability of contrail effects of individual flights.
  [Lines 21-24]: Estimates of the effective radiative forcing of aviation from 1940 to 2018 indicate that a significant part (best estimate 2/3) stems from non-CO2 effects, including contrails, NOx and $H_2O$ emissions (Lee et al., 2021). However, this estimated 2/3 contribution is a global and annual average and especially the contribution from contrails varies significantly between individual flights (Grewe et al., 2014; Dahlmann et al., 2021; Teoh et al., 2022).

3. *Lines 25-32: Can the relevance of the CCFs to this particular study please be explained here? Is the connection that this study is testing reliability of the model for future studies using the CCF approach?*

- While this work is independent of the CCF development and application, it adds to the evaluation of uncertainties on the CCF calculation. We have added some lines to put this work more into perspective of the CCF application.
  [Line 35-43]: This study aims to provide additional information on model uncertainties and their effects on the resulting CCFs. Additionally, the results from this study will serve as a basis for recalculating contrail-specific CCFs, further enhancing climate-optimized flight planning . As discussed by Grewe et al. (2017), the use of CCFs in daily operations requires the implementation of fast algorithms in daily weather forecast systems, and their corresponding mitigation potential  must be evaluated in a consistent modelling framework.  Such a framework ensures that the derived climate responses to aviation emissions are physically and methodologically coherent, allowing a fair comparison across different scenarios and emission types. Achieving this requires an integrated setup that includes the algorithmic CCFs as well as an air traffic optimizer integrated into a general circulation model.

4. *Line 32: Isn't NOx emission on ozone quite outside the focus of this study?*

- Thank you for raising this point. This is an example for CCF validation approaches that had been conducted in the past. We tried to make this fact clearer.
  [Line 43-45]: So far, this approach has only been applied to NOx emission effects, and a validation exercise has been carried out to test the impact of NOx emissions on ozone (Yin et al., 2023; Rao et al., 2022).

5. *Lines 95-7: Why no unnudged version? The model presumably on its own has different biases than the nudged version. Do the authors know if the the RH biases would be worse without the nudging, or if the T bias is affected by (or is largely a result of) the nudging?*

- Thank you for the comment. This is discussed in detail by Jöckel et al. 2016. Yes, indeed the un-nudged model shows both, a temperature and a humidity bias. Both are reduced with the standard nudging procedure, which usually does not involve a temperature bias correction, since the wave-zero (in spectral space of the model), i.e. the mean of temperature is not nudged. In case, wave-zero of the temperature is included in the nudging, the temperature bias is, as to be expected, considerably reduced, and, as we show here, also effecting the humidity bias.

  [Line 123-128]: While no fully unnudged simulation was conducted for this study, previous work by Jöckel et al. (2016) has shown that the unnudged EMAC model exhibits both temperature and humidity biases. Standard nudging procedures (excluding the global mean temperature or wave-zero term) already help reduce these biases compared to the completely free-running model state. Including the global mean temperature in the nudging (i.e., using MTN setups) further reduces the temperature bias, and as a result also influences the humidity fields as shown in this study.

6. *Line 104-126: I'd expected this model description would focus more on contrails. Can it be mentioned here that the contrail submodels will be further discussed in Section 2.2.? Also, can the relevance of the described methane tracer and CH4 sub-model please be made clear upfront? These are both relevant because they are factors for stratospheric water vapor, right?*

- Thank you for pointing this out. We will clarify the relevance of the methane tracer and the CH4 sub-model in the revised manuscript. Specifically, methane oxidation in the stratosphere is a key source of water vapour, which influences the background humidity conditions relevant for contrail formation and persistence. By accounting for methane's contribution to stratospheric water vapour through the CH4 sub-model, we ensure that the simulated atmospheric composition—and therefore the conditions for contrail development—are more realistic. We will add a brief explanation highlighting this importance in the relevant section. We also have restructured the contrail section so that all information is now presented together in a more compact form, and also included a reference to Section 2.2 when discussing the Contrail Submodule.

  [Line 129-137]: As in RD1SD-base-01, the time evolution of the methane tracer at the lower model boundary was prescribed by Newtonian relaxation towards the data provided by CCMI-2. Since methane oxidation in the stratosphere constitutes a significant source of water vapour, accurately representing its variation is crucial for realistic atmospheric humidity distributions. In the $CH_4$ sub-model (Winterstein and Jöckel, 2021), the stratospheric water vapour contribution from methane oxidation  is calculated using photolysis rates provided online by the photo-chemistry sub-model JVAL (Riede et al. 2009). This ensures that the background humidity conditions, which influence contrail formation and persistence, are more accurately represented.

  Two EMAC sub-models are  particularly relevant for this study: S4D : ('sampling in 4 dimensions' (Jöckel et al., 2010)): and CONTRAIL (Version 1.0; Frömming et al., 2014). : The CONTRAIL sub-model  identifies regions where contrails and contrail cirrus may form

variables are calculated following Burkhardt et al. (2008). With the and/or persist, as described in more detail in Section 2.2.

7. *Lines 101-2: By "global mean temperature", is this nudging performed independently at each model level? Should be specified.*

- The global mean temperature (wave zero of temperature) meant in this context is model level dependent. We add this information in the revised manuscript.
[Line 120-122]: A stronger interference in the case of the MTN simulations (involving additional nudging of the global mean temperature at the corresponding model levels) results in a generally better agreement of EMAC model temperature with ERA5 reanalysis data.

8. *Lines 107 and 129: Can the Materials and Methods section please be edited to make clear the relationships among CONTRAIL, PotCov (which is not currently mentioned here), SAC, and ISSR. Also, S4D seems to be quite separate, so I find the references to "two sub-models" and "two approaches" for separate pairs quite confusing, given how the second of each list is the same but not the first, if I correctly understand that CONTRAIL is where SAC is calculated.*

- Thank you for pointing this out. We made the relationship between Contrail, SAC and ISSR clearer and the potcov calculation is explained in more detail. We rearrange the paragraph so that each sub-model is introduced and described immediately after it is mentioned.
[Line 135-137]: Two EMAC sub-models are  particularly relevant for this study: S4D  ": ('sampling in 4 dimensions' (Jöckel et al., 2010)): and CONTRAIL (Version 1.0; Frömming et al., 2014). : The CONTRAIL sub-model  identifies regions where contrails and contrail cirrus may form  and/or persist, as described in more detail in Section 2.2. The S4D sub-model  enables online sampling of model data along aircraft flight trajectories
[Line 157-185]: In this study, we investigate two different approaches to  determine atmospheric regions capable of forming persistent contrails and contrail cirrus, without distinguishing between linear contrails and contrail cirrus.  One approach uses the parametrizations developed by Burkhardt et al. (2008) and Burkhardt and Kärcher (2009), as implemented in the EMAC CONTRAIL sub-model, to estimate the potential contrail coverage (PotCov). PotCov represents the fraction of a grid box that can be maximally covered by contrails under given large-scale temperature and humidity conditions. It is computed as the difference between (i) the maximum possible combined coverage of natural cirrus clouds and contrails, and (ii) the coverage of natural cirrus clouds alone. Both coverage values depend on critical temperature and humidity thresholds with respect to ice, which are adapted for large-scale model conditions. These thresholds also incorporate the Schmidt–Appleman Criterion (SAC; Schumann, 1996), which is derived from fundamental physical principles of mass, momentum, and energy conservation. In essence, the SAC determines whether hot exhaust gases mixing with cold ambient air will become supersaturated with respect to supercooled liquid water, enabling ice particle nucleation. This is often visualized using a 'mixing line' in

the temperature–humidity space, which tracks how temperature and humidity evolve as exhaust and ambient air combine. Besides ambient temperature and humidity, the SAC accounts for aircraft-related parameters such as propulsion efficiency and fuel combustion heat. In this study, we adopt engine parameters as provided by Grewe et al. (2014). Contrails persist only when the atmosphere is supersaturated with respect to ice; otherwise, they dissipate shortly after formation. Because these small-scale processes cannot be explicitly resolved in EMAC, their effects are represented through the parametrizations following Burkhardt et al. (2008). Further details on the calculation of PotCov are given in Grewe et al. (2014) and Frömming et al. (2021). Reanalysis data, such as ERA5, do not provide fields for PotCov directly. Therefore, an alternative approach is needed to compare the atmospheric potential for contrail persistence. This alternative identifies ice-supersaturated regions (ISSRs)  following the methodology suggested by Dietmüller et al. (2023).  ISSRs are usually defined by the following two criteria: The temperature must be below 235 K to separate from mixed-phase regions (Pruppacher et al., 1998) and the relative humidity with respect to ice (RHice) must exceed 100% (see  Reutter et al., 2020). However, when studying the sub-grid-scale variability in the relative humidity field of numerical weather forecast model data, such as ERA5, it is essential to consider RHice thresholds below 100% as demonstrated by Irvine et al. (2014). Dietmüller et al. (2023) explored different RHice thresholds for ERA5 and conducted a comparison with MOZAIC data (Petzold et al., 2020) in the European region.  They found the best agreement between ERA5 and observations is achieved when the RHice threshold is set to 90%. ~~Alternatively, the atmospheric ability to form persistent contrails and contrail cirrus in EMAC can be calculated at each time step according to Burkhardt et al. (2008) and Burkhardt and Kärcher (2009).Here, the fraction of a grid box available for contrail coverage is determined by the Schmidt-Appleman criterion, incorporating local temperature and humidity conditions conducive to contrail formation. This fraction is calculated as the difference between the potential coverage of contrails and cirrus clouds combined and the coverage of natural cirrus clouds alone. For our study, we utilise values for the overall propulsion efficiency of an aircraft and the fuel combustion heat, which are essential for calculating the SAC, from Schumann et al. (2000). Further details about the calculation method for the potential contrail coverage are given by Frömming et al. (2021) and Grewe et al. (2014).~~ In this study we explore RHice thresholds between 90% and 100%. Here, we compare both approaches using identical EMAC atmospheric data under various model setups, as described in Section 3.2.

9. *Lines 125-6: Isn't the NAFC interesting for more prominent reasons, e.g. a sizable share of global flights and contrail occurrence?*

- Yes, there are more reasons why the NAFC is of particular interest. We mention them now in the manuscript.
  [Line 153-154]: The NAFC is of particular interest due to its high share of global air traffic, frequent occurrence of contrail formation conditions, and because CCFs have been calculated  in this region (Frömming et al., 2021).

10. *Lines 129-30: Could this explanation of the two contrail formation approaches please specify upfront how these are similar and differ? The ice microphysical criteria appear overall similar, but the Schmidt-Appleman approach factors in jet engine characteristics, if I understand this correctly. Would a reasonable interpretation be that the second approach is more advanced and expected to be more accurate?*

- Thank you for pointing this out. We have revised the text to clarify the differences between the two approaches. Both methods identify conditions favorable for ice crystal formation, but the Schmidt–Appleman-based approach (used in EMAC) explicitly incorporates aircraft and engine parameters—such as propulsion efficiency and fuel combustion heat—into the calculation. In contrast, the ice-supersaturation approach relies solely on atmospheric humidity and temperature conditions. As a result, the Schmidt–Appleman-based method can be considered more physically comprehensive and may yield more accurate estimates of contrail formation potential. We have updated the manuscript to highlight these distinctions in the discussion section. The restructure of the method section (see Comment #8) should also help to explain the two contrail formation approaches better.
[Line 601-607]: Contrail formation can be identified through two main approaches: one based solely on ice-supersaturated regions (ISSRs) and another employing the SAC to derive potential contrail coverage (PotCov). Both require similar microphysical conditions (temperature and humidity thresholds), but the SAC-based approach is more physically comprehensive as it includes aircraft engine parameters such as propulsion efficiency and specific fuel combustion heat (Schumann, 1996; Schumann et al., 2000; Grewe et al., 2014). While the simpler ISSR-based method can identify where contrails may form, the SAC-based approach is generally considered more refined and potentially more accurate, as it directly links aircraft properties and ambient conditions.

11. *Lines 144-6: What jet characteristics were factored into CONTRAILS? The cited Schumann 2000 paper mentions a number of propulsion efficiencies. Also, can the authors please comment on whether they suspect the results of this study would be quite different if alternative parameters had been tested?*

- We refer now to Grewe 2014, where the numbers used for this study are listed in Table 3. We also discuss the impact of alternative parameter in the discussion section now.
[Line 170-171]: In this study, we adopt engine parameters as provided by Grewe et al. (2014).
[Line 607- 611]: Consequently, varying these aircraft parameters can influence the spatial extent and frequency of predicted contrails. For instance, adjusting the propulsion efficiency would have a relatively small effect on contrail formation regions, while changes to parameters like the emitted water vapour could more substantially alter the predicted contrail coverage. As demonstrated in frameworks like CoCiP (Schumann, 2012), an accurate representation of engine and fuel parameters can significantly improve the realism of contrail predictions, and thus, inform more effective climate mitigation strategies in aviation.

12. *Lines 216-7: Is there also shape criteria here? How are contrails distinguished from other clouds?*

- This kind of RGBs enhances linear contrail properties, in particular the fact that contrails are optically thin ice clouds composed of small ice crystals. These properties induce brightness temperature differences in the sensor channels used here, i.e. the brightness temperatures at 8.7 µm minus the brightness temperatures at 10.8 µm and the brightness temperatures at 10.8 µm minus the brightness temperatures at 12.0 µm. Thus, contrails are emphasized in these false colour composites. Their linear shape is of course crucial in order to distinguish natural thin cirrus with small ice crystals from the anthropogenic contrails. All these criteria are used in automatic contrail detection, either based on image processing (Mannstein et al. (1999)) or on machine learning (Meijer et al. (2022), Ng et al. (2024)). However, in this study we don't apply any automatic contrail detection since the focus is not on single contrails, but do a visual inspection of the satellite RGBs in order to identify contrail areas that can be compared to the model. This has been explained now in more details in the manuscript. [Line 267-270]: Here, contrails appear as dark blue or black  linear objects and are easy to identify This kind of pictures is often used  as a basis for automatic contrail detection as e.g. in (Meijer et al., 2022). In this study, we restrict ourselves to a visual analysis of the ash RGBs in order to identify areas of frequent contrail formation.

13. *Lines 291-2: ERA5 does not calculate contrail occurrence, right? For another model to have "exhibited a larger contrail formation region compared to ERA5" suggests otherwise.*

- Thank you for pointing that out. We changed it to regions with ice-supersaturation. [Line 341-343]: Due to the differences in temperature and humidity between the model and ERA5 reanalysis data, all EMAC setups generally exhibited  larger ice-supersaturated region compared to ERA5.

14. *Lines 307-9: Is "PotCov" simply a direct result of the Schmidt-Appleman criteria described in section 2.2? If so, it should be named in Section 2.2 and not first mentioned here.*

- PotCov is now introduced in Section 2.2. "PotCov" is not a direct result of the Schmidt-Apple criteria but the SAC is also used during the calculation of PotCov. We now provide reference to an article that explains the concept in more detail. Also see answer so to review comment #8.
  [Line 159-160]: One approach uses the parametrizations developed by Burkhardt et al. (2008) and Burkhardt and Kärcher (2009), as implemented in the EMAC CONTRAIL sub-model, to estimate the potential contrail coverage (PotCov).

15. *Lines 354: Do these "low humidity values" matter? These would never form contrails, presumably, so it's confusing for this to be commented on without this clarification.*

- We changed the order and highlight that these "low humidity values" are not important for contrails.
  [Line 425-435]: The analysis of water vapour mixing ratios across various model setups and measurement data, illustrated in Figure 6, reveals

 notable discrepancies between model results and observations. For high levels of humidity ranging between 100 and 800 ppmV, predominantly present in the upper troposphere, the results from the standard temperature nudging simulations persistently provide lower values than the HALO FISH and SHARC measurements (see Figure 6, left). This "dry bias" is strongly reduced in simulations with mean temperature nudging (see Figure 6, right). This could be directly connected to the temperature bias reduction found in Figure 2, as temperature influences humidity. In contrast, for low humidity values between 5 to 10 ppmV, which are located in the lower stratosphere, the model consistently overpredicts compared to the observations by up to six times across all model setups. Nevertheless, these low humidity values are not relevant for contrail formation. This model "wet bias" appears with both nudging methods.

16. *Lines 368-9: Is it correct to say that the model results are being used to test for ice-supersaturated regions (ISSRs) as described in Section 2.2? I find the current description of "we focus on ice-supersaturated conditions, where the ambient relative humidity with respect to ice (RHice) exceeds 100%" not sufficiently clear how this connects to the methods described earlier.*

- Yes, it is correct, we investigate the depends of the diagnosed areas on the used model resolution. We make it clearer that this is the approach explained in Section 2.2.
  [Line 440-442]: We analyse the point-by-point correlation between measurement data and different model setups for  atmospheric conditions when contrails can persist longer than a few minutes and significantly impact the climate, following the approach of Gierens et al. (2020). We  use the approach discussed in Section 2.2 to identify ice-supersaturated  regions.

17. *Line 372: I don't understand the word "gradually" here. Is the threshold actually evolving over time, or this is meant to convey "moderately"?*

- We changed the word to stepwise. For the diagnosed categorical statistical measures, the threshold is not time dependent and we repeat the diagnose 11 time.
  [Line 445-446]:
  We explore relative humidity thresholds (Dietmüller et al. (2023)), by stepwise evaluating categorical statistical measures for model thresholds between 100 to 90% (Fig. 7).

18. *Fig. 8: In the lowest panel (PotCov) there are substantial differences between simulations. Can the authors please comment on whether they think this is due to the resolution differences being important or is simply noise?*

- Thank you for raising this point. The observed differences in potential contrail coverage among the various model setups largely arise from the nature of the contrail cover

parameterization itself. This parameterization inherently represents a fraction of a grid box that can be occupied by contrails, and varying the spatial resolution effectively changes how this fraction is sampled and represented. As a result, some of the differences are indeed influenced by changes in resolution rather than reflecting true meteorological variability.

In other words, the "contrail cover concept" can introduce a form of noise or apparent variation that cannot be solely attributed to atmospheric conditions. While higher resolution simulations may produce more spatially heterogeneous patterns, this initial analysis does not allow us to conclusively determine whether these differences are systematically driven by resolution or primarily result from the parameterization and sampling effects. Further dedicated studies and sensitivity tests would be required to clarify the root causes of these observed differences.

[Line 525-530]: Differences in potential contrail coverage between higher and lower resolution setups are partly a consequence of the underlying "contrail cover concept." This parametrization, designed to represent partial cloud cover within discrete model grid boxes, can lead to variability that does not solely reflect atmospheric conditions. Instead, it introduces a form of "noise" where changes in resolution and sampling can alter the fraction of the grid box potentially filled with contrails. From this initial analysis, it is not possible to definitively conclude the extent to which these differences stem from systematic resolution effects or from such parametric variability.

19. *Fig. 9 caption and Line 466: Two instances of "L31T63" that should be "T63L31".*

- Thank you for pointing that out. We changed the order for these two instances.
  [Fig 9]: The maximum potential contrail coverage between 300 and 200 hPa (3 model levels) derived from the EMAC  T63L31 STN model setup.
  [Line 533-536]: The maximum  PotCov in the T63L31 STN model indicates the highest value observed between 300 and 200 hPa (bottom panel in Fig. 9).

20. *Lines 476-497: The first 20 or so lines of the Discussion are really Summary. This looks to me more appropriate for the "Summary and Conclusions" section that follows.*

- We followed this suggestion and moved the first part of the discussion to the summary and conclusions part.

21. *Line 550: I don't see any reference to a higher ETS score to back this statement. Is this based on some analysis that is not shown in the study? This should be stated in the text.*

- As this observation results from our data analysis we moved this text to the result section. We have deleted the sentence in the discussion section. The new text is provided below.
  [Line 472-476]: By reducing in our analysis both the model and observational relative humidity thresholds below 90%, more data points are counted as contrail-forming conditions. This increases the number of hits and improves the ETS score to approximately

0.5. However, this artificial enhancement in agreement compromises physical realism, as contrail formation typically requires conditions near full saturation.

22. *Lines 552-5: I don't see any reference to the SAC approach in the test for skill (last sub-section of Section 4), which seems to be based entirely on the ISSR approach, so I find this statement confusing.*

- Thank you for highlighting this point. The mention of the Schmidt-Appleman criterion (SAC) in that particular sentence refers specifically to how EMAC calculates potential contrail coverage (PotCov). We revised the text to emphasize that SAC underlies the contrail coverage calculation within EMAC, not the skill test in Section 4. The skill assessment there is indeed based on the ISSR approach. Therefore, the reference to SAC in this part of the discussion is intended to clarify the foundation of the PotCov output rather than describe the method used in the final skill test. We will further clarify this distinction in the manuscript. [Line 649-651]: Since the Schmidt-Appleman criterion, a thermodynamic theory that has been thoroughly tested and validated, is the basis for  the PotCov calculation in EMAC, any deviations in contrail coverage mainly arise from inaccurate representations of key parameters for contrail formation, such as temperature or humidity.

**Reference:**

Jöckel, P., Tost, H., Pozzer, A., Kunze, M., Kirner, O., Brenninkmeijer, C. A. M., Brinkop, S., Cai, D. S., Dyroff, C., Eckstein, J., Frank, F., Garny, H., Gottschaldt, K.-D., Graf, P., Grewe, V., Kerkweg, A., Kern, B., Matthes, S., Mertens, M., Meul, S., Neumaier, M., Nützel, M., Oberländer-Hayn, S., Ruhnke, R., Runde, T., Sander, R., Scharffe, D., and Zahn, A.: Earth System Chemistry integrated Modelling (ESCiMo) with the Modular Earth Submodel System (MESSy) version 2.51, Geoscientific Model Development, 9, 1153–1200, https://doi.org/10.5194/gmd-9-1153-2016, 2016.

Grewe, V., Frömming, C., Matthes, S., Brinkop, S., Ponater, M., Dietmüller, S., Jöckel, P., Garny, H., Tsati, E., Dahlmann, K., Søvde, O. A., Fuglestvedt, J. S., Berntsen, T., Shine, K. P., Irvine, E. A., Champougny, T., and Hullah, P.: Aircraft routing with minimal climate impact: the REACT4C climate cost function modelling approach (V1.0), Geoscientific Model Development, 7, 175–201, https://doi.org/10.5194/gmd-7-175-2014, 2014.

Meijer, V. R., Kulik, L., Eastham, S. D., Allroggen, F., Speth, R. L., Karaman, S., and Barrett, S. R. H.: Contrail coverage over the United States before and during the COVID-19 pandemic, Environmental Research Letters, 17, 034 039, https://doi.org/10.1088/1748-9326/ac26f0, 2022.

Mannstein H, Meyer R and Wendling P 1999 Operational detection of contrails from NOAA-AVHRR-data Int. J. Remote Sens. 20 1641–60

J.Y.H. Ng et al. Contrail detection on goes-16 abi with the opencontrails dataset. IEEE Trans. Geosci. Remote. Sens., 2023

Grewe, V., Dahlmann, K., Flink, J., Frömming, C., Ghosh, R., Gierens, K., Heller, R., Hendricks, J., Jöckel, P., Kaufmann, S. H. E., Kölker, K., Linke, F., Luchkova, T., Lührs, B., Van Manen, J., Matthes, S., Minikin, A., Niklaß, M., Plohr, M., Righi, M., Rosanka, S., Schmitt, A. R., Schumann, U., Unterstraßer,

S., Vázquez-Navarro, M., Voigt, C., Wicke, K., Yamashita, H., Zahn, A., and Ziereis, H.: Mitigating the Climate Impact from Aviation: Achievements and Results of the DLR WeCare Project, Aerospace, 4, 34, https://doi.org/10.3390/aerospace4030034, 2017.

Stenke, A., Grewe, V., and Ponater, M.: Lagrangian transport of water vapor and cloud water in the ECHAM4 GCM and its impact on the cold bias, Climate Dynamics, 31, 491–506, https://doi.org/10.1007/s00382-007-0347-5, 2007.

Charlesworth, E., Plöger, F., Birner, T., Baikhadzhaev, R., Ábalos, M., Abraham, N. L., Akiyoshi, H., Bekki, S., Dennison, F., Jöckel, P., Keeble, J., Kinnison, D. E., Morgenstern, O., Plummer, D. A., Rozanov, E., Strode, S. A., Zeng, G., Egorova, T., and Riese, M.: Stratospheric water vapor affecting atmospheric circulation, Nature Communications, 14, https://doi.org/10.1038/s41467-023-39559-2, 2023.

Jöckel, P., Kerkweg, A., Pozzer, A., Sander, R., Tost, H., Riede, H., Baumgaertner, A., Gromov, S., and Kern, B.: Development cycle 2 of the Modular Earth Submodel System (MESSy2), Geoscientific Model Development, 3, 717–752, https://doi.org/10.5194/gmd-3-717-2010, 2010.

Jöckel, P., Tost, H., Pozzer, A., Kunze, M., Kirner, O., Brenninkmeijer, C. A. M., Brinkop, S., Cai, D. S., Dyroff, C., Eckstein, J., Frank, F., Garny, H., Gottschaldt, K.-D., Graf, P., Grewe, V., Kerkweg, A., Kern, B., Matthes, S., Mertens, M., Meul, S., Neumaier, M., Nützel, M., Oberländer-Hayn, S., Ruhnke, R., Runde, T., Sander, R., Scharffe, D., and Zahn, A.: Earth System Chemistry integrated Modelling (ESCiMo) with the Modular Earth Submodel System (MESSy) version 2.51, Geoscientific Model Development, 9, 1153–1200, https://doi.org/10.5194/gmd-9-1153-2016, 2016.

Wang, Z., Bugliaro, L., Jurkat-Witschas, T., Heller, R., Burkhardt, U., Ziereis, H., Dekoutsidis, G., Wirth, M., Groß, S., Kirschler, S., Kaufmann, S., and Voigt, C.: Observations of microphysical properties and radiative effects of a contrail cirrus outbreak over the North Atlantic, Atmospheric Chemistry and Physics, 23, 1941–1961, https://doi.org/10.5194/acp-23-1941-2023, 2023

Voigt, C., Schumann, U., Minikin, A., Abdelmonem, A., Afchine, A., Borrmann, S., Boettcher, M., Buchholz, B., Bugliaro, L., Costa, A., Curtius, J., Dollner, M., Dörnbrack, A., Dreiling, V., Ebert, V., Ehrlich, A., Fix, A., Forster, L., Frank, F., Fütterer, D., Giez, A., Graf, K., Grooß, J., Groß, S., Heimerl, K., Heinold, B., Hüneke, T., Järvinen, E., Jurkat, T., Kaufmann, S. H. E., Kenntner, M., Klingebiel, M., Klimach, T., Kohl, R., Krämer, M., Krisna, T. C., Luebke, A., Mayer, B., Mertes, S., Molleker, S., Petzold, A., Pfeilsticker, K., Port, M., Rapp, M., Reutter, P., Rolf, C., Rose, D., Sauer, D., Schäfler, A., Schlage, R., Schnaiter, M., Schneider, J., Spelten, N., Spichtinger, P., Stock, P., Walser, A., Weigel, R., Weinzierl, B., Wendisch, M., Werner, F., Wernli, H., Wirth, M., Zahn, A., Ziereis, H., and Zöger, M.: ML-CIRRUS: The Airborne Experiment on Natural Cirrus and Contrail Cirrus with the High-Altitude Long-Range Research Aircraft HALO, Bulletin of the American Meteorological Society, 98, 271–288, https://doi.org/10.1175/bams-d-15-00213.1, 2017.

Li, Y., Mahnke, C., Rohs, S., Bundke, U., Spelten, N., Dekoutsidis, G., Groß, S., Voigt, C., Schumann, U., Petzold, A., and Krämer, M.: Upper-tropospheric slightly ice-subsaturated regions: frequency of occurrence and statistical evidence for the appearance of contrail cirrus, Atmospheric Chemistry and Physics, 23, 2251–2271, https://doi.org/10.5194/acp-23-2251-2023, 2023.